# Caveolin-1 protects endothelial cells from extensive expansion of transcellular tunnel by stiffening the plasma membrane

Camille Morel[1], Eline Lemerle[2], Feng-Ching Tsai[3], Thomas Obadia[4,5],
Nishit Srivastava[6], Maud Marechal[1], Audrey Salles[7], Marvin Albert[8],
Caroline Stefani[9], Yvonne Benito[10], François Vandenesch[11], Christophe Lamaze[12],
Stéphane Vassilopoulos[2], Matthieu Piel[6], Patricia Bassereau[3],
David Gonzalez-Rodriguez[13], Cecile Leduc[14]*, Emmanuel Lemichez[1]*

[1]Institut Pasteur, Université Paris Cité, CNRS UMR6047, Inserm U1306, Unité des Toxines Bactériennes, Département de Microbiologie, Paris, France; [2]Sorbonne Université, INSERM UMR974, Institut de Myologie, Centre de Recherche en Myologie, Paris, France; [3]Institut Curie, PSL Research University, CNRS UMR168, Physics of Cells and Cancer Laboratory, Paris, France; [4]Institut Pasteur, Université Paris Cité, Bioinformatics and Biostatistics Hub, Paris, France; [5]Institut Pasteur, Université Paris Cité, G5 Infectious Diseases Epidemiology and Analytics, Paris, France; [6]Institut Curie and Institut Pierre Gilles de Gennes, PSL Research University, Sorbonne University, Paris, France; [7]Institut Pasteur, Université Paris Cité, Photonic Bio-Imaging, Centre de Ressources et Recherches Technologiques (UTechS-PBI, C2RT), Paris, France; [8]Institut Pasteur, Université Paris Cité, Image Analysis Hub, Paris, France; [9]Benaroya Research Institute at Virginia Mason, Department of Immunology, Seattle, United States; [10]Centre National de Référence des Staphylocoques, Hospices Civiles de Lyon, Lyon, France; [11]CIRI, Centre International de Recherche en Infectiologie, Université de Lyon, Inserm U1111, Université Claude Bernard Lyon 1, CNRS UMR5308, ENS de Lyon, Lyon, France, Lyon, France; [12]Institut Curie, PSL Research University, INSERM U1143, CNRS UMR3666, Membrane Mechanics and Dynamics of Intracellular Signaling Laboratory, Paris, France; [13]Université de Lorraine, LCP-A2MC, Metz, France; [14]Université Paris Cité, Institut Jacques Monod, CNRS UMR7592, Paris, France

*For correspondence:
cecile.leduc@ijm.fr (CL);
elemiche@pasteur.fr (EL)

**Abstract** Large transcellular pores elicited by bacterial mono-ADP-ribosyltransferase (mART) exotoxins inhibiting the small RhoA GTPase compromise the endothelial barrier. Recent advances in biophysical modeling point toward membrane tension and bending rigidity as the minimal set of mechanical parameters determining the nucleation and maximal size of transendothelial cell macroaperture (TEM) tunnels induced by bacterial RhoA-targeting mART exotoxins. We report that cellular depletion of caveolin-1, the membrane-embedded building block of caveolae, and depletion of cavin-1, the master regulator of caveolae invaginations, increase the number of TEMs per cell. The enhanced occurrence of TEM nucleation events correlates with a reduction in cell height due to the increase in cell spreading and decrease in cell volume, which, together with the disruption of RhoA-driven F-actin meshwork, favor membrane apposition for TEM nucleation. Strikingly, caveolin-1 specifically controls the opening speed of TEMs, leading to their dramatic 5.4-fold larger

widening. Consistent with the increase in TEM density and width in siCAV1 cells, we record a higher lethality in CAV1 KO mice subjected to a catalytically active mART exotoxin targeting RhoA during staphylococcal bloodstream infection. Combined theoretical modeling with independent biophysical measurements of plasma membrane bending rigidity points toward a specific contribution of caveolin-1 to membrane stiffening in addition to the role of cavin-1/caveolin-1-dependent caveolae in the control of membrane tension homeostasis.

## eLife assessment

This **important** study identifies the role of caveolin-1 and cavin1 as regulators of transendothelial macroaperture (TEM). The methodology used is rigorous and **compelling**, and further research can point to more mechanistic understanding of the process.

## Introduction

The endothelial cell monolayer lining the inner surface of the vascular tree forms an active semi-permeable barrier that actively responds to hemodynamic forces to control blood pressure (*Aird, 2007a*; *Aird, 2007b*). Regulation of the endothelial barrier involves a bidirectional interplay between actomyosin cytoskeleton and plasma membrane mechanics, which together control paracellular exchanges at cell–cell junctions (*Komarova et al., 2017*), as well as transcellular exchanges through less-characterized transcellular openings (*Aird, 2007a*; *Aird, 2007b*; *Guo et al., 2016*). Defining how membrane mechanical parameters affect the dynamics of opening and widening of transendothelial pores will help better define the role of these openings both physiological and pathophysiological processes.

Large transcellular pores in the endothelium can contribute to aqueous humor outflow in the eyes, can form during the diapedesis of leukocytes, or appear when endothelial cells are intoxicated by several bacterial toxins known to breach the endothelial barrier (*Lemichez et al., 2013*; *Braakman et al., 2015*; *Barzilai et al., 2017*). Bacterial mono-ADP-ribosyltransferases (mART) exotoxins, which catalyze the post-translational modification of the small GTPase RhoA for inhibition, induce the spontaneous formation of transendothelial cell macroaperture (TEM) tunnels up to 10–20 μm in diameter (*Boyer et al., 2006*; *Gonzalez-Rodriguez et al., 2020*; *Maddugoda et al., 2011*). This group of mART includes the exotoxin C3 (ExoC3) from *Clostridium botulinum* and the epidermal differentiation inhibitors (EDIN) A, B, and C close homologs from *Staphylococcus aureus* (*Sugai et al., 1992*; *Chardin et al., 1989*; *Paterson et al., 1990*). The intravenous injection of EDIN or ExoC3 triggers vascular leakage (*Boyer et al., 2006*; *Rolando et al., 2009*). Cumulative evidence also indicates that EDIN favors the translocation of *S. aureus* through vascular tissues for dissemination (*Munro et al., 2010*; *Courjon et al., 2015*). Although TEM tunnels have not yet been visualized *in vivo*, they form *ex vivo* in the endothelium lining the inner surface of rat aortas either infected with *S. aureus* producing the RhoA-targeting EDIN mART or treated with EDIN (*Boyer et al., 2006*). Mechanistically, the inhibition of RhoA signaling by bacterial mART exotoxins disrupts contractile actomyosin stress fibers, which otherwise stiffen in response to shear forces exerted by the blood flow that tends to compress cells (*Boyer et al., 2006*; *Gonzalez-Rodriguez et al., 2020*; *Maddugoda et al., 2011*; *Sugai et al., 1992*; *Chardin et al., 1989*). In line with this, compressive mechanical forces applied to the apical region of endothelial cells bring into close contact the apical and basal plasma membranes to nucleate transcellular pores (*Ng et al., 2017*). Although force-induced nucleation of TEMs can be triggered in the absence of inhibition of RhoA, the magnitude of the force required to mechanically open TEMs dramatically decreases when RhoA is inhibited (*Ng et al., 2017*). This led to postulate that TEMs form because of a disruption of actomyosin stress fibers under the control of RhoA, which concur to reduce the height of cells as they spread out (*Boyer et al., 2006*; *Tsai et al., 2022*). The physical mechanisms that set both the nucleation of TEMs and their maximal width remain to be ascertained experimentally.

TEMs represent a remarkable model system to identify the molecular machinery that controls cell mechanics (*Gonzalez-Rodriguez et al., 2020*). Our current view of the mechanical forces at play in the opening of TEMs largely arises from the phenomenological analogy drawn between TEM tunnel opening and the formation of dry patches in viscous liquid forced to spread on a solid surface, a physical phenomenon referred to as liquid dewetting (*Gonzalez-Rodriguez et al., 2012*). Notably,

the analogy drawn between liquid and cellular dewetting suggests that the increase in membrane tension in response to cell spreading is the driving force of TEM nucleation also contributing to define the initial speed of opening, that is, before that membrane tension relaxation around TEM edges and build-up of an actomyosin bundle encircling TEMs counteract the opening (*Maddugoda et al., 2011*; *Gonzalez-Rodriguez et al., 2012*; *Stefani et al., 2017*). Of note, the high curvature of the plasma membrane at the edge of TEMs also points toward a contribution of membrane bending rigidity, that is, the energy required to locally bend the membrane surface, in their dynamics. Experimental evidence is still required to ascertain the roles of membrane tension and bending rigidity in the dynamics of TEM tunnel nucleation and enlargement.

Caveolae are mechano-regulators of the plasma membrane that spontaneously flatten to accommodate acute mechanical loads and thereby prevent membrane rupturing (*Sinha et al., 2011*; *Parton et al., 2020a*). Caveolin oligomers are essential components of caveolae pits (*Parton et al., 2006*; *Porta et al., 2022*). Caveolin-1 homo-oligomers form flat discoid complexes embedded into the inner leaflet of the membrane (*Porta et al., 2022*). The process of invagination of caveolar pits of 50–100 nm diameter involves the association of caveolin-1 with cavin-1 cytosolic structural protein (also referred to as PTRF) (*Ludwig et al., 2013*; *Ariotti et al., 2015*; *Ludwig et al., 2016*; *Yang and Scarlata, 2017*; *Lamaze et al., 2017*). Caveolae pits are further stabilized at their neck by the assembly of a ring of dynamin-like EHD2 oligomers (*Daumke et al., 2007*; *Yeow et al., 2017*; *Nishimura and Suetsugu, 2022*). In endothelial cells, the crosstalk between caveolin-1 and RhoA during actomyosin contractility leads to critical mechano-sensing and adaptation to hemodynamic forces (*Yu et al., 2006*; *Del Pozo et al., 2021*). For example, caveolin-1-deficient mice undergo an increase in the endothelial nitric oxide (eNOS) signaling that is critical to activate RhoA signaling via nitration-mediated inhibition of the GTPase-activating protein (GAP) activity of p190RhoGAP-A (*Siddiqui et al., 2011*; *Razani et al., 2001*; *Miyawaki-Shimizu et al., 2006*). Cell biology and epidemiological studies have started to establish the importance of caveolin-1 in host–pathogen interactions (*Machado et al., 2012*; *Spaan et al., 2022*; *Medina et al., 2006*). Considering the protective function of caveolae membrane invaginations in ensuring membrane tension homeostasis in response to acute mechanical loads (*Sinha et al., 2011*; *Dewulf et al., 2019*), caveolin-1 and cavin-1 represent potential mechano-regulators of transcellular pore dynamics.

Using cutting-edge microscopy techniques, we establish that endothelial cells treated with RhoA-targeting mART display caveolae pits at the ventral part of the plasma membrane and a loose F-actin meshwork that frame large cytosolic zones devoid of F-actin. We report that the disruption of caveolae and depletion of caveolin-1, using siRNA-targeting cavin-1 or caveolin-1, respectively, increases the number of TEM per cell. We establish that the increase in TEM opening correlates with a decrease in cell volume concomitant with cell spreading, which concur to a decrease in cell height. Strikingly, we establish that caveolin-1, contrary to cavin-1, controls the widening of TEMs, which rules out a role of caveolae membrane invaginations. This increase in both the number and width of TEMs correlates with a higher susceptibility of caveolin-1-deficient mice to the lethal effect of EDIN-B mART activity during *S. aureus* septicemia. Finally, using an independent biophysical technique and theoretical modeling, we provide compelling evidence that caveolin-1 contributes, independently of cavin-1, to increasing plasma membrane bending rigidity. These data point toward a role of cavin-1/caveolin-1 components of caveolae in the regulation of TEM nucleation through the modulation of membrane tension. Hence, we ascribe to caveolin-1 a function in stiffening the plasma membrane that is independent from cavin-1-dependent caveolae, which eventually modulates TEMs opening speed and width.

## Results

### Caveolae control the density of TEM tunnels

The formation of large TEM tunnels is observed in endothelial cells intoxicated with EDIN-like factors from *S. aureus*, as well as with the prototypic exoenzyme C3 (ExoC3) from *C. botulinum*. Considering that RhoA inhibition increases endothelial cell spreading, we investigated the impact of ExoC3 treatment on the distribution and density of caveolae in the inner leaflet of the plasma membrane in human umbilical vein endothelial cells (HUVECs) (*Figure 1A–C*). We used a cell unroofing approach that allows direct visualization of membrane invaginations on a platinum replica via transmission electron microscopy (*Lemerle et al., 2023*). In both naïve and ExoC3-treated cells, we observed

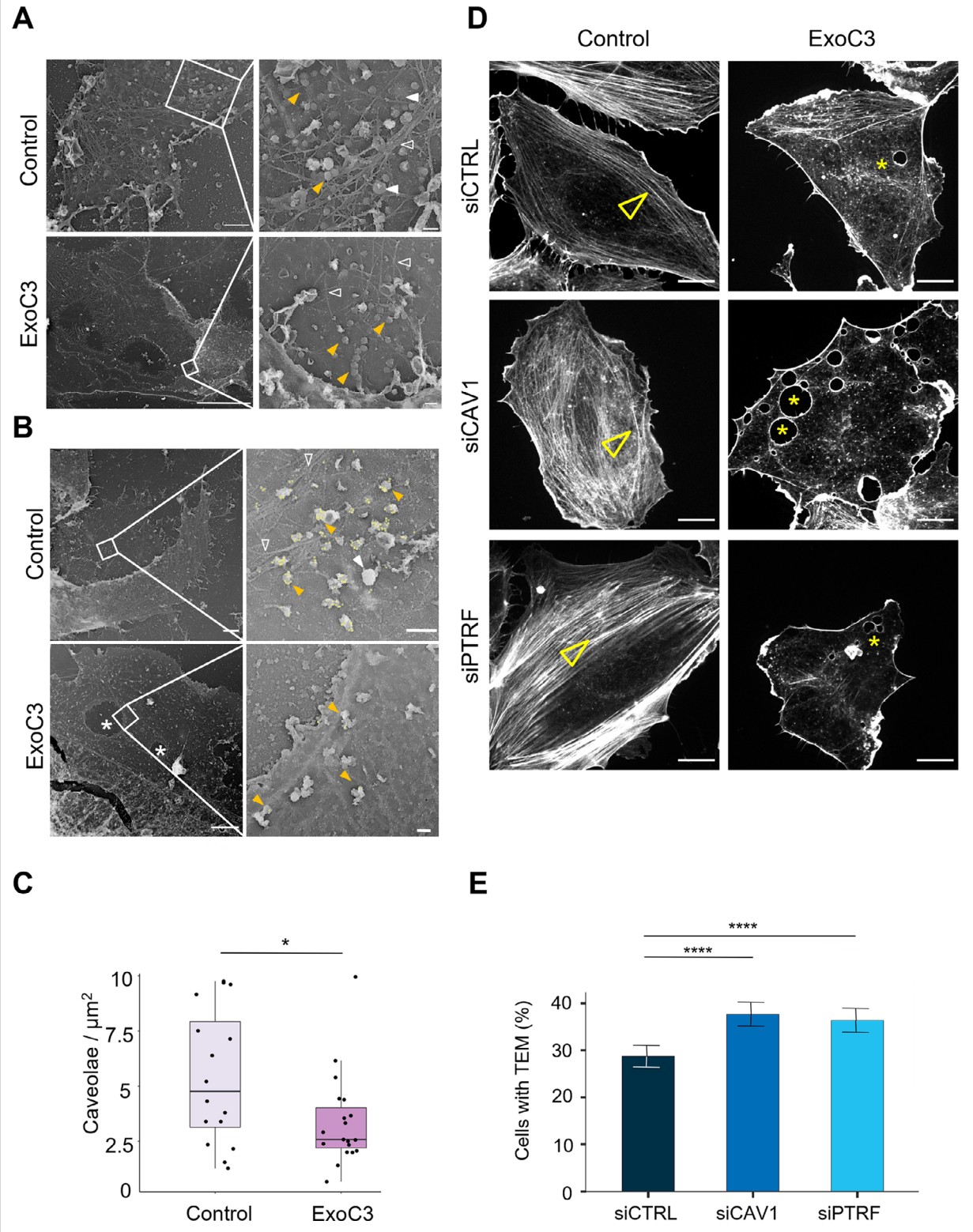

**Figure 1.** RhoA inhibition decreases the density of caveolae and actin stress fibers. (**A, B**) Transmission electron micrographs show unroofed human umbilical vein endothelial cells (HUVECs) that were either left untreated (control) or treated with 100 μg/ml of ExoC3 (ExoC3) for 24 hr. Right panels show membrane areas at higher magnification with (i) invaginated caveolae (yellow arrowhead), (ii) clathrin-coated pits and patches (plain white arrowhead), (iii) actin filaments (empty white arrowhead), and (iv) transendothelial cell macroaperture (TEM) tunnels in ExoC3-treated cells (white star). Scale bars left panels: 1 μm for control and 5 μm for ExoC3 condition with 200 nm higher magnifications on right panels. (**B**) Yellow dots show immunogold-labeled

*Figure 1 continued on next page*

*Figure 1 continued*

GFP-CAV1. (**C**) Boxplot shows the density of caveolae per µm² of plasma determined on electron micrographs. Values analyzed with a mixed-effects linear model with random intercept and Tukey's correction for pairwise comparison. *p=0.027 (n = 16 cells from three technical replicates). (**D**) Confocal spinning disk images show F-actin cytoskeleton of HUVECs left untreated (control) or treated with 100 µg/ml ExoC3 (ExoC3) overnight after 24 hr of transfection with siRNA control (siCTRL), targeting caveolin-1 (siCAV1) or cavin-1 (siPTRF). Cells were stained with phalloidin-TRITC. Arrowheads show stress fibers and stars show transcellular tunnels bounded by F-actin. Scale bars, 20 µm. (**E**) Histograms show the percentages of ExoC3-treated cells displaying at least one TEM (n = 1400 cells, eight independent experiments). Error bars show normal asymptotic 95% confidence intervals (CI). Data analysis with mixed-effect logistic regression model with correction for multiple comparisons using Tukey's honestly significant difference (HSD) test. ****p<0.0001, for both siCTRL vs. siCAV1 and siCTRL vs. siPTRF conditions with no difference between siCAV1 and siPTRF conditions.

The online version of this article includes the following source data and figure supplement(s) for figure 1:

**Source data 1.** Data for *Figure 1C and E*.

**Figure supplement 1.** Controls of siRNAs and RhoA ADP-ribosylation efficacies.

**Figure supplement 1—source data 1.** Files with western blots.

**Figure supplement 1—source data 2.** Data for western blot quantification in *Figure 1—figure supplement 1B and C*.

**Figure supplement 2.** Caveolin-1 and cavin-1/PTRF depletion increases transendothelial cell macroaperture (TEM) formation in both ExoC3- and epidermal differentiation inhibitor (EDIN)-treated cells.

**Figure supplement 2—source data 1.** Data for *Figure 1—figure supplement 2A–C*.

**Figure supplement 3.** Depletion of the caveolar accessory component EHD2 increases transendothelial cell macroaperture (TEM) formation.

**Figure supplement 3—source data 1.** Files with western blots.

**Figure supplement 3—source data 2.** Data for *Figure 1—figure supplement 3B and C*.

---

membrane-associated cytoskeleton filaments, the honeycomb structure of clathrin lattices, clathrin-coated pits, and rough aspects of pit-like caveolar invaginations (*Figure 1A*). We verified that the treatment of cells with ExoC3 had no impact on caveolin-1 and cavin-1 expression (*Figure 1—figure supplement 1A*). Of note, the density of the cytoskeletal filaments was less pronounced in the ExoC3-treated cells (*Figure 1D*, arrowhead). Cells were transfected with a plasmid encoding GFP-caveolin-1 to confirm by immunogold labelling the presence of caveolin-1-positive caveolae invaginations at the ventral side of the plasma membrane (*Figure 1B*). We quantified the density of caveolae in ExoC3-treated and control cells. In total, we analyzed an area $A_{area}$ > 175 µm² under each condition (n ≥ 16 membrane areas of independent cells per condition). We recorded a 1.6-fold decrease in the mean density of caveolae in the ExoC3-treated cells, which exhibited 3.4 ± 2.1 caveolae/µm² of membrane, compared to 5.4 ± 3.1 caveolae/µm² of membrane in untreated cells (*Figure 1C*). The pool of plasma membrane-located caveolae remaining in ExoC3-treated cells is evenly distributed with no detectable accumulation around TEM tunnels (*Figure 1B*). ExoC3 treatment reduces the density of invaginated caveolae at the plasma membrane, pointing toward the interest of studying the impact of caveolae depletion in TEM formation.

Before functional analysis of the impact of caveolae components on TEM tunnel nucleation and growth, we analyzed the previously established stoichiometry of caveolin-1 and cavin-1 in response to siRNA treatments (*Hill et al., 2008*). We observed that small interfering RNA (siRNA)-mediated depletion of caveolin-1 (siCAV1) strongly reduces both caveolin-1 and the cellular pool of cavin-1/PTRF by approximately 80% (*Figure 1—figure supplement 1B*). In contrast, about one-half of the pool of caveolin-1 remained

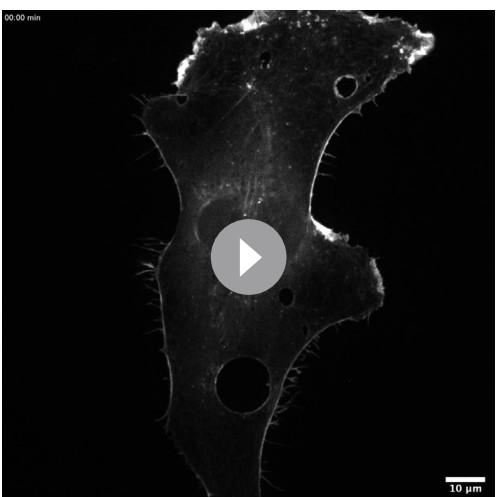

**Video 1.** Cycles of transendothelial cell macroaperture (TEM) tunnel opening and closure in GFP-LifeAct-expressing human umbilical vein endothelial cell (HUVEC) transfected with a control siRNA prior to intoxication with ExoC3 at 100 µg/ml for 24 hr. Video recorded at one frame every 15 s and played at seven images per second. Scale bar, 10 µm.

https://elifesciences.org/articles/92078/figures#video1

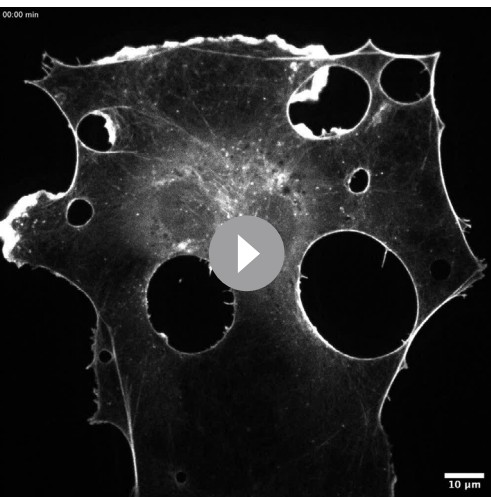

**Video 2.** Cycles of transendothelial cell macroaperture (TEM) tunnel opening and closure in GFP-LifeAct-expressing human umbilical vein endothelial cell (HUVEC) transfected with siRNA targeting caveolin-1 prior to intoxication with ExoC3 at 100 µg/ml for 24 hr. Video recorded at one frame every 23 s and played at seven images per second. Scale bar, 10 µm.
https://elifesciences.org/articles/92078/figures#video2

unaffected by the siRNA-targeted depletion of cavin-1/PTRF (siPRTF) (*Figure 1—figure supplement 1B*). We concluded that siPTRF leaves a cellular pool of caveolin-1 unaffected, contrary to siCAV1. Both siRNA treatments had no effect on the extent of mono-ADPribosylation of RhoA (*Figure 1—figure supplement 1C*). We next quantified the efficiency of TEM tunnel formation in different conditions of siRNA treatment. HUVECs were treated with siCTRL, siCAV1, or siPTRF prior to ExoC3 treatment. Examples of cells treated in these conditions are shown in *Figure 1D* and *Videos 1–3*. Quantitative measurements showed a higher percentage of cells displaying at least one TEM in the population of caveolin-1 or cavin-1-deficient cells (37.6 ± 1.3% siCAV1 and 36.4 ± 1.3% siPTRF vs. 28.8 ± 1.2% siCTRL) (*Figure 1E*). Moreover, we observed a significant increase in the density of TEMs per cell in siCAV1 ($N_{TEM\_siCAV1}$ = 1.96) and siPTRF ($N_{TEM\_siPTRF}$ = 1.36) conditions compared to siCTRL ($N_{siCTRL}$ = 0.91) (*Figure 1—figure supplement 2A*). We verified that such an increase in TEM density per cell and within the cell population was also induced by the ExoC3-like RhoA-targeting mART EDIN from *S. aureus* (*Figure 1—figure supplement 2B and C*) or by siRNA-targeting EHD2 caveolae-stabilizing protein, a treatment that leaves caveolin-1 and cavin-1 unaffected (*Figure 1—figure supplement 3*). Together, these data show that the efficiency of the formation of TEMs is inversely correlated to the density and stability of caveolae at the plasma membrane.

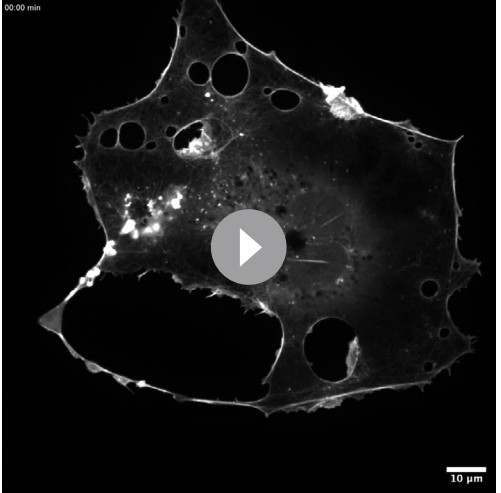

**Video 3.** Cycles of transendothelial cell macroaperture (TEM) tunnel opening and closure in GFP-LifeAct-expressing human umbilical vein endothelial cell (HUVEC) transfected with siRNA targeting cavin-1 prior to intoxication with ExoC3 at 100 µg/ml for 24 hr. Video recorded at one frame every 20.5 s and played at seven images per second. Scale bar, 10 µm.
https://elifesciences.org/articles/92078/figures#video3

## Caveolae buffer the decrease in cell height triggered by RhoA inhibition-driven spreading

The formation of TEMs due to the disruption of RhoA-driven F-actin polymerization and actomyosin contraction is facilitated by a reduction in cell height that promotes contact between the apical and basal membranes (*Ng et al., 2017; Tsai et al., 2022*). We first quantified the mesh size of F-actin network using 2D stochastic optical reconstruction microscopy (STORM) that integrates F-actin signals with an ~30 nm resolution over the whole-cell thickness, which can be as thin as 50 nm at the cell periphery (*Maddugoda et al., 2011*). The siCTRL, siCAV1, and siPTRF-transfected HUVECs displayed a classical organization of actin cytoskeleton with actin stress fibers intertwined with a high-density meshwork of F-actin (*Figure 2A*, control). When siRNA-treated cells were next intoxicated with ExoC3, we observed the disruption of actin stress fibers together with a reorganization of the actin cytoskeleton into a loosely intertwined and irregular F-actin meshwork defining large zones devoid of

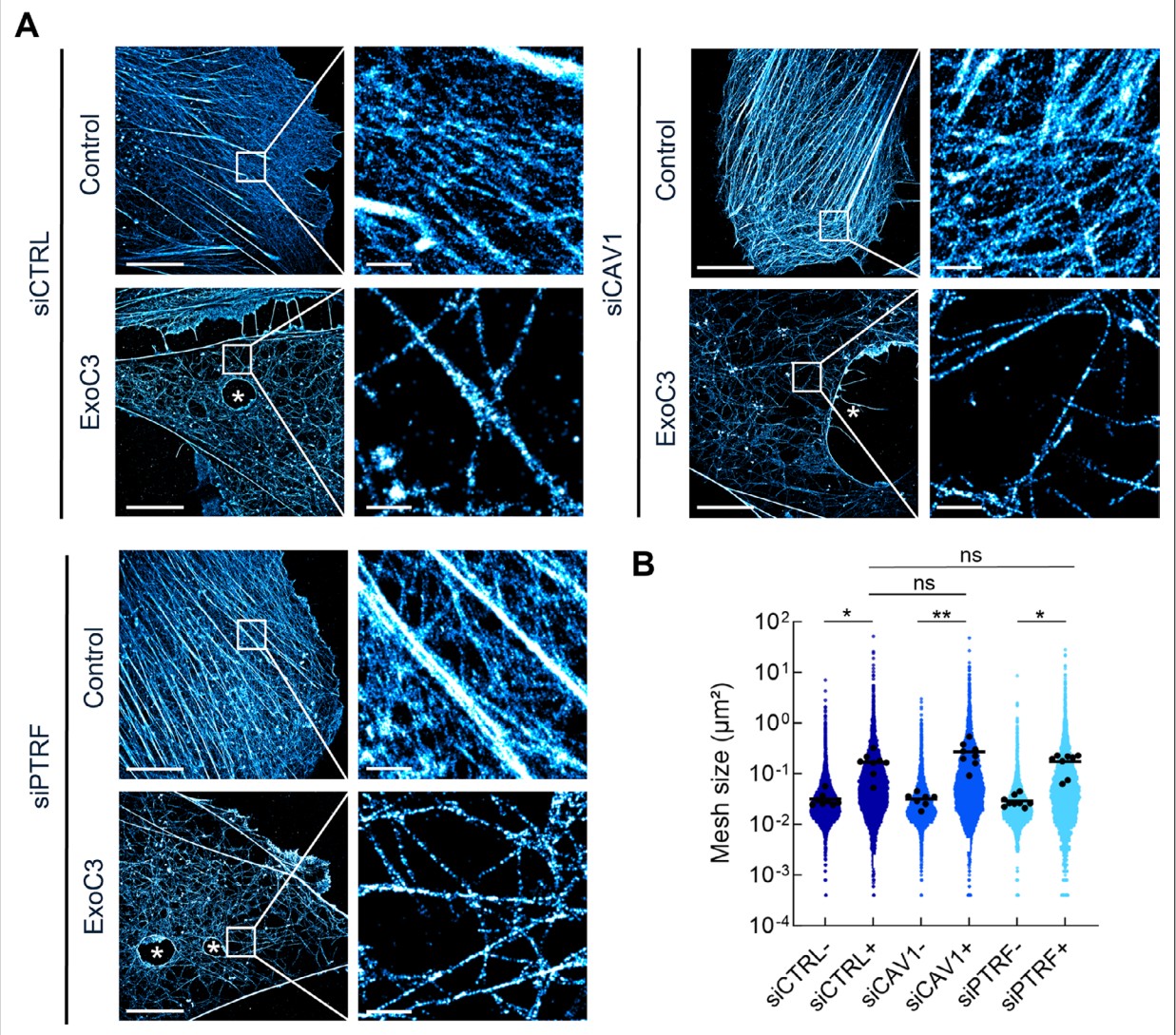

**Figure 2.** RhoA inhibition decreases F-actin mesh size with no significant effect of caveolae components. (**A**) 2D stochastic optical reconstruction microscopy (STORM) images show the disruption of actin bundles and intertwined F-actin in cells intoxicated with ExoC3 at 100 µg/ml for 24 hr. Human umbilical vein endothelial cells (HUVECs) either left untreated or treated with ExoC3 before F-actin staining with phalloidin-AF647. Scale bar, 10 µm. High-magnification images are shown in the right panels. Scale bar, 1 µm. (**B**) Scatter dot plot shows the average mesh size per cell (µm²) of the F-actin network (black dots) overlaid with all individual values of mesh size (blue dots) shown in logarithmic scale. Quantification was performed with 2D STORM images of control (-) and ExoC3-treated cells (+) that were first treated with siCTRL (dark blue), siCAV1 (blue), and siPTRF (light blue). **p<0.01, *p<0.05 calculated with a nested *t*-test (n = 8–9 cells per group, three independent replicates).

The online version of this article includes the following source data for figure 2:

**Source data 1.** Data for *Figure 2*.

F-actin (*Figure 2A*, ExoC3). In all conditions of siRNA treatment, we have measured that ExoC3 treatment induces a significant increase in actin mesh size of about 0.2 µm² compared to non-intoxicated cells (*Figure 2B*). No significant difference of mesh size was recorded between siCTRL, siCAV1, or siPTRF conditions before and after treatment with ExoC3 (*Figure 2B*). Indeed, we observed in cells treated with ExoC3 no specific cellular pattern or bimodal distribution of mesh size between the different siRNA conditions but a rather very heterogeneous distribution of mesh size values that could reach a few square microns in all conditions. We concluded that in addition to the well-established impact of RhoA inhibition on actin stress fibers it generates a loose F-actin meshwork framing large cellular zones devoid of F-actin of same extent in all siRNA conditions.

**Table 1.** Cell spreading area, volume, and height.

The means and standard deviations (SDs) of the cell spreading area (*Figure 3C*) and volume (*Figure 3D*). Cell height was estimated by the ratio between the cell volume and cell area, and the SD was estimated via error propagation.

| Conditions | Mean cell area $A$ (µm²) | SD of the cell area (µm²) | Mean cell volume $V$ (µm³) | SD of the cell volume (µm³) | Cell height $h$ (µm) | SD of the height (µm) |
|---|---|---|---|---|---|---|
| siCTRL- | 6583 | 5168 | 2320 | 3147 | 0.35 | 0.75 |
| siCTRL+ | 9025 | 7258 | 2710 | 4326 | 0.30 | 0.72 |
| siCAV1- | 7048 | 5583 | 2146 | 3214 | 0.30 | 0.70 |
| siCAV1+ | 9831 | 7984 | 2357 | 3010 | 0.24 | 0.50 |
| siPTRF- | 7539 | 5450 | 1914 | 2234 | 0.25 | 0.48 |
| siPTRF+ | 10,139 | 7873 | 2282 | 2628 | 0.23 | 0.43 |

We next investigated the role played by caveolin-1 and cavin-1 in the control of the cell height ($h$), an essential parameter in TEM nucleation. The mean value of this cell shape parameter is proportional to the ratio between the volume ($V$) and spreading area ($A$) of cells (*Table 1*). The spreading area was estimated from the measurement of cell contours (*Figure 3A*). The volume of cells was quantified by fluorescent dye exclusion, as depicted in *Figure 3B*, and as previously described (*Zlotek-Zlotkiewicz et al., 2015*). Upon intoxication of siCTRL-treated cells, we recorded a significant increase of 36% of the spreading area together with 17% increase of the cell volume (*Figure 3C and D* and *Table 1*). The dominant impact of cell area changes over the volume parameter accounts for the calculated decrease of cell height of 15% (*Table 1*). Hence, when siCAV1- and siPTRF-transfected cells were treated with ExoC3, we recorded a reduction in cell volume concomitant to the increase in their spreading area (*Figure 3C and D* and *Table 1*). The reinforced reduction in cell height recorded in siCAV1 or siPTRF-treated cells is in good agreement with our findings showing that these treatments increase the density of TEMs. Note that in the absence of ExoC3 we recorded that the effects of siPTRF on cell shape parameters were slightly stronger than those of siCAV1, thereby highlighting the key involvement of caveolae in the control of cell height (*Figure 3C and D*). The depletion of caveolae and the inhibition of RhoA work together to reduce cell height that favors together with the loss of F-actin the nucleation of TEMs.

## Caveolin-1 and cavin-1 have a different impact on TEM expansion dynamics

Since the TEM opening is transient, we next characterized the impact of caveolin-1 and cavin-1 in TEM dynamics, which allows us to define TEM maximal width. We implemented a pipeline of live-cell imaging and semi-automatic analysis that relies on the detection of LifeAct-GFP signal around TEMs to segment the opening area (*Tsai et al., 2022*). *Figure 4A* shows the projections of the first image recorded for each TEM that formed over 1 hr recording (*Figure 4A*). We see that TEM tunnels preferentially open at the cell periphery where the cell height is minimal (*Figure 4A*). We then measured the maximal values of each TEM areas over 1 hr of video recording (*Figure 4B*), as well as the speeds of opening and closure (*Figure 4C*). Interestingly, we recorded 5.4-fold wider TEMs in siCAV1-treated cells compared to siPTRF- and siCTRL-treatments (*Figure 4B* and *Videos 1–3*). Interestingly, the time for TEMs to reach their maximal diameter ($t_{max}$) and the overall TEM cycles ($t_c$) remains identical in all conditions (*Figure 4C*). Thus, TEMs became wider in siCAV1 condition in the absence of change in the time frame of opening, suggesting that caveolin-1 controls the speed of opening. To capture initial values of TEM opening speed ($S_o$), we used a 15-fold higher temporal resolution, that is, using a video recording of one image per second. Strikingly, we recorded a twofold higher opening speed in caveolin-1-depleted cells: $S_{o\_siCAV1} = 2.4$ µm²/s versus $S_{o\_siCTRL} = 1.1$ µm²/s in control cells, which contrasts with the minor impact of siPTRF with $S_{o\_siPTRF} = 1.7$ µm²/s (*Figure 4D*). In conclusion, we have identified a specific function of caveolin-1 in the regulation of the opening speed of TEMs that has a critical impact on TEM maximal width.

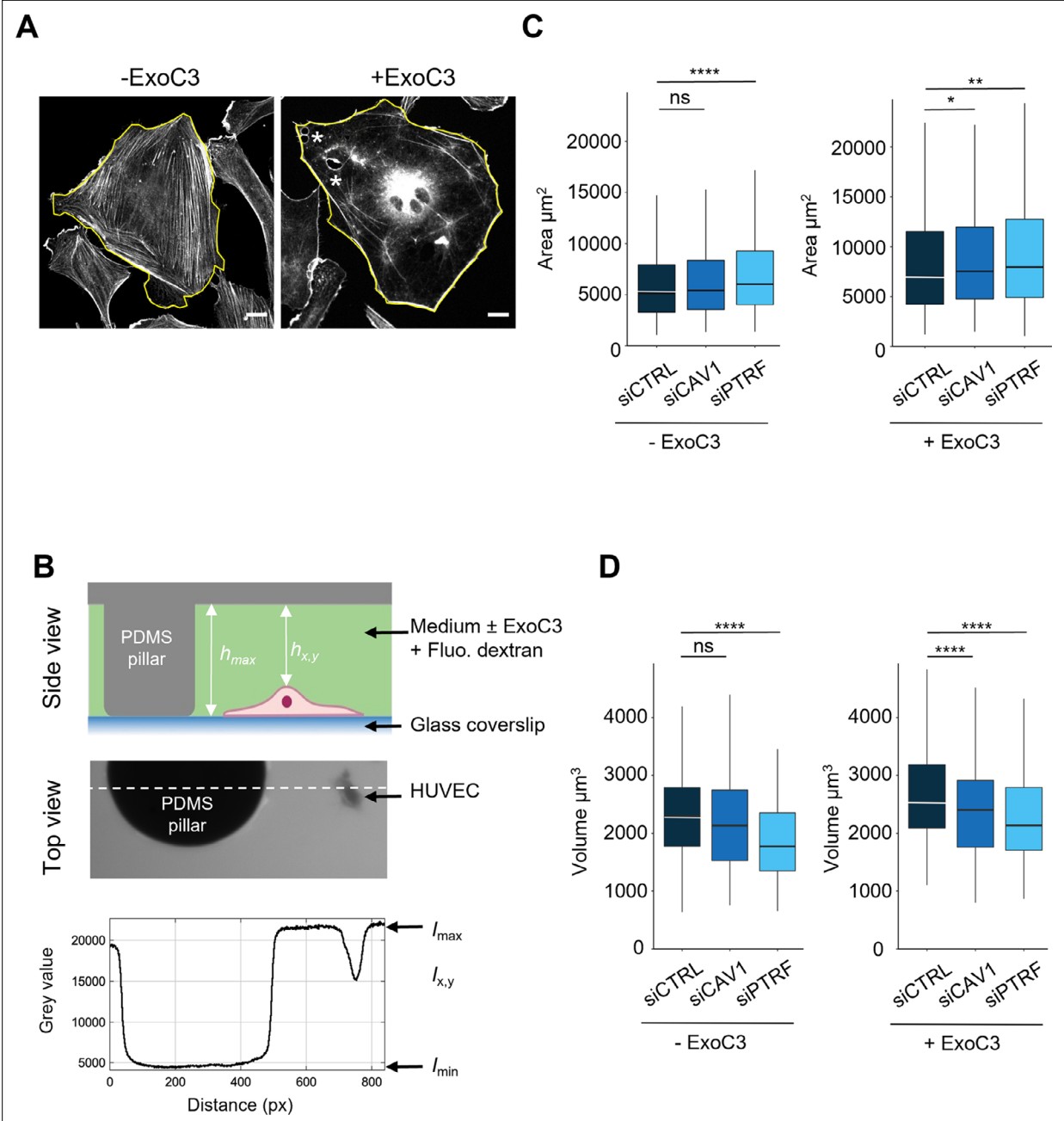

**Figure 3.** Regulation of human umbilical vein endothelial cell (HUVEC) area and volume by cavin-1/PTRF, caveolin-1, and RhoA. (**A**) Confocal spinning disk images of HUVECs stained with phalloidin-FITC and selected perimeters (yellow line) in the absence (- ExoC3) or presence of ExoC3 (+ExoC3). Stars show the presence of transendothelial cell macroaperture (TEM) in ExoC3-treated cells. Scale bar, 20 μm. (**B**) Schematic representation of the microfluidic chamber used to measure cell volume by fluorescence exclusion. Briefly, from the top (i) side view of the chamber in which a cell adheres to a coverslip. The PDMS pillar sustains the ceiling (gray), and the maximal height of the chamber $h_{max}$ (background) is known. The siCTRL, siCAV1, or siPTRF-transfected HUVECs were seeded in the chamber and remained either untreated or treated with ExoC3. High molecular weight dextran-FITC (green) was added to the chamber and is nonpermeant to cells (values $h_{x,y}$); (ii) raw epifluorescence image showing a typical field of HUVEC; and (iii) the graph of fluorescence intensities (in grayscale) shows the function of distance along the dotted line. Parameters $I_{max}$ and $I_{min}$ yield values of maximum and minimum fluorescence intensities. Values for cell volume ($V_{cell}$) were obtained by integrating the fluorescence intensities $h_{max} - h_{x,y}$ over the cell area. (**C**) Boxplots show the distribution of TEM area values estimated from measures of their perimeters, which is shown in (**A**). Measurements were performed with HUVEC transfected with siCTRL (dark blue), siCAV1 (blue), or siPTRF (light blue) and then treated with ExoC3 (+ExoC3) or untreated (- ExoC3). Measurements were performed with n > 698 untreated cells and n > 595 treated cells, five independent experiments. (**D**) Boxplots show the distribution of cell volumes, as described in (**B**). Measurements were performed on HUVEC transfected with siCTRL, siCAV1, or siPTRF and then treated with ExoC3 (+ExoC3) or untreated (-ExoC3). Data are from n = 216 and n = 308 cells after siCTRL ± ExoC3 treatment (dark blue), n = 197 and

*Figure 3 continued on next page*

Figure 3 continued

n = 266 cells after siCAV1 ± ExoC3 treatment (blue) and n = 152 and n = 157 cells after siPTRF 10 ± ExoC3 treatment (light blue); three independent experiments. The graphs show technical replicates pooled together. The data were analyzed with a mixed-effect generalized linear model with Gamma log-link function, random intercept accounting for technical variability, and Tukey's correction for pairwise comparisons between control and each siRNA treatment, ****$p<0.0001$, **$p<0.01$, *$p<0.05$ and ns, not significant.

The online version of this article includes the following source data for figure 3:

**Source data 1.** Data for *Figure 3C and D*.

## Loss of caveolin-1 sensitizes mice to EDIN-B lethal effects during *S. aureus* sepsis

We reasoned that the increased density and width of TEMs triggered by the cellular depletion of caveolin-1 should result in higher susceptibility of caveolin-1-deficient (*Cav1-/-*) mice to the inhibition of RhoA by EDIN-like mART exotoxins during bloodstream infection by *S. aureus*. This would establish caveolin-1 as a resistance factor in the pathogenicity triggered by EDIN-like factors from *S. aureus*. To address this question, we first established the susceptibility of *Cav1-/-* mice and wildtype littermates (*Cav1+/+*) to *S. aureus* strain LUG1799 (WT *edinB*) in a model of septicemia (*Figure 5A*). LUG1799 belongs to the European lineage ST80 derived from community-acquired methicillin-resistant *S. aureus*, which expresses EDIN-B mART exotoxin (EDIN-B) (*Courjon et al., 2015*). We first monitored over a period of 7 d the survival of *Cav1-/-* and *Cav1+/+* mice challenged with three increasing *inocula* of WT *edinB*. All the mice, except one *Cav1-/-* mouse, recovered from infection triggered with $5 \times 10^6$ CFU of bacteria/mouse (CFU/m), while all mice died 24 hr after intravenous injection of $5 \times 10^8$ CFU/m. In contrast, when mice were challenged with an intermediate dose of WT *edinB* of $5 \times 10^7$ CFU/m, we recorded a higher lethality of *Cav1-/-* compared to *Cav1+/+* mice (p<0.001, Mantel–Cox test). All the *Cav1-/-* mice died on day 1 post-challenge, whereas the death of *Cav1+/+* occurred with a delay of 1–2 d, with one *Cav1+/+* mouse surviving (*Figure 5A*, $5 \times 10^7$ CFU/m). The loss of caveolin-1 expression thus sensitizes mice to bloodstream infection triggered by *S. aureus*-produced EDIN-B. We then explored the pathogenic effect of EDIN-B mART activity of RhoA. In control experiments with *Cav1+/+* mice infected with either WT *edinB* or Δ*edinB* strains, we saw no difference of susceptibility between the strains and despite a broad range of four different *inocula* tested (*Figure 5—figure supplement 1*). In this set of experiments, we defined values of lethal dose 50 (LD$_{50}$) at $2.5 \times 10^7$ CFU/m with both strains of *S. aureus*. When we challenged *Cav1-/-* mice with this LD$_{50}$ of *S. aureus* ($2.5 \times 10^7$ CFU/m), we saw a lower survival with *S. aureus* WT *edinB* (n = 11 death/18 mice) compared to Δ*edinB* (n = 4 death/17 mice) (p=0.0123, Mantel–Cox test) (*Figure 5C*). To provide a causal link between the pathogenicity of EDIN-B recorded in *Cav1-/-* mice and the inhibition of RhoA catalyzed by EDIN-B, we performed a similar experiment with *S. aureus* Δ*edinB* complemented with a plasmid expressing either WT EDIN-B (Δ*edinB* pEDIN-B) or the catalytically inactive mutant EDIN-B-RE (Δ*edinB* pEDIN-B-RE). We saw a significant decrease in survival in caveolin-1-deficient mice infected with Δ*edinB* pEDIN-B compared to mice infected with Δ*edinB* pEDIN-B-RE. We concluded that in the context of bloodstream infection by *S. aureus*, caveolin-1 expression confers a resistance to the pathogenicity triggered by EDIN-B mART activity on RhoA.

## Differential impacts of caveolin-1 and cavin-1 on membrane rigidity

Since we expect that $S_o$ depends on membrane mechanical parameters (see the physical model description in 'Materials and methods'), this suggests that caveolin-1, but not cavin-1, modulates TEM opening speed by affecting membrane mechanics. We then inferred changes in membrane mechanical properties upon depletion of caveolin-1 or cavin-1 using a refined theoretical model of cellular dewetting. This model accounts for the presence of several TEMs opening simultaneously (see the physical model description in 'Materials and methods'). Importantly, the model is based on the hypothesis that plasma membrane deformation, enabling TEM nucleation and tunnel growth, is a function of membrane tension and bending rigidity, assuming a limited contribution of membrane adhesion to cortical cytoskeleton (*Helfrich, 1973*). Indeed, consistent with this hypothesis of a minimal impact of cortical cytoskeleton elements on membrane mechanics, the 2D STORM images (*Figure 2A*) established a massive disruption of the dense F-actin meshwork in ExoC3-treated cells, leaving large cellular zones devoid of F-actin. As explained in 'Materials and methods' in detail, the

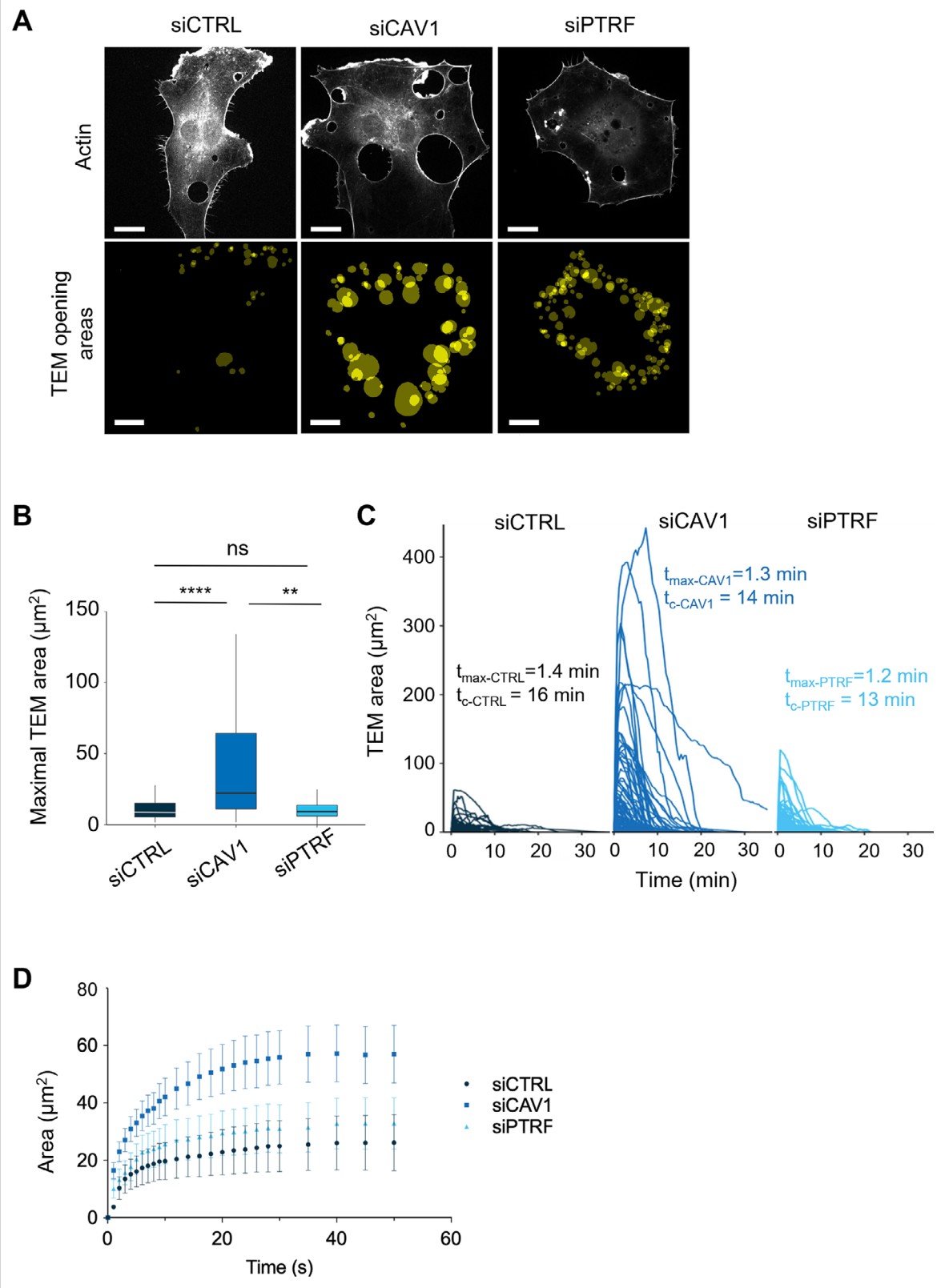

**Figure 4.** Caveolin-1 controls the transendothelial cell macroaperture (TEM) opening speed and maximum size. (**A**) Images show examples of projections of all tunnels upon TEM initial opening (lower panel) in human umbilical vein endothelial cells (HUVECs) transfected with LifeAct-GFP expression plasmid and siCTRL, siCAV1, or siPTRF captured during 1 hr of live imaging. LifeAct-GFP HUVECs transfected with different siRNAs were treated with ExoC3 and recorded by live imaging for 1 hr. All initial TEM opening was based on the first frame in which TEM tunnels formed using ICY.

*Figure 4 continued on next page*

*Figure 4 continued*

The lower panel shows the projection of cumulative areas of initial TEM opening identified during 1 hr of live imaging. Scale bars, 20 µm. (**B**) Boxplot shows the distribution of TEMs, the maximal and median area values in HUVECs cotransfected with LifeAct-GFP-expressing plasmid and siCTRL, siCAV1, or siPTRF prior ExoC3 treatment. Maximal areas were determined based on each kinetic parameter of TEM dynamics, as shown in (**C**). The data represent n > 105 TEMs in seven cells of each treatment group from >3 independent experiments. Graph shows technical replicates pooled together. Statistical data analysis using a mixed-effect generalized linear model with Gamma log-link function, random intercept, and Tukey's correction for multiple comparisons. ****p<0.0001, **p<0.01, and ns, nonsignificant. (**C**) The graph shows variations in TEM areas as a function of time expressed in minutes. HUVECs transfected with siCTRL, siCAV1, or siPTRF were treated with ExoC3 for 24 hr. The calculated values of $t_{max}$ that corresponded to the time of opening to the time when the maximal areas were observed and the values of $t_c$ corresponded to the time frame of a complete cycle of opening and closing are indicated on the graph for each condition. The data are from n > 105 TEMs of seven cells per treatment from >3 independent experiments. (**D**) Graph shows variations in mean values, expressed in seconds, in the TEM areas of cells treated with ExoC3. The curves were plotted with data obtained from time-lapse video recorded at one frame per second for 30 min. LifeAct-GFP-expressing cells transfected with siCTRL, siCAV1, and siPTRF. The data correspond to n > 22 TEMs per condition from four independent experiments.

The online version of this article includes the following source data for figure 4:

**Source data 1.** Data for *Figure 4B–D*.

model predicts that the initial opening speed $S_o$ is proportional to the membrane tension, consistent with a critical role of membrane tension in TEM growth (*Gonzalez-Rodriguez et al., 2020*). From our measurements, we deduce that cells treated with siCAV1 have a significant twofold higher membrane tension than the controls, and no effect in siPTRF-treated cells (see 'Materials and methods'). Moreover, according to the model, the maximum TEM size $R_{max}$ depends strongly on membrane bending rigidity but weakly on the initial membrane tension. Thus, from our $R_{max}$ measurements, the model allows the estimation of the bending rigidities $\kappa$ in the different conditions. Using the following values for the siCTRL-treated cells – an initial membrane tension $\sigma_0 = 2.5\ 10^{-5}$ N/m (*Raucher and Sheetz, 2000*) and a bending rigidity $\kappa_{siCTRL} = 23.4\ k_BT$ (direct measurement, see below) – the model predicts a 55% reduction in membrane rigidity from a loss of expression of caveolin-1 and a lower reduction of 15% upon a loss of cavin-1 expression.

To test these predictions, we have treated cells with methyl-β-cyclodextrin to deplete cholesterol from the plasma membrane and reduce its bending rigidity (*Steinkühler et al., 2019*); unfortunately, this treatment affected the cell morphology, which precluded further analysis. Therefore, we used an independent approach to measure the membrane bending rigidity in different conditions of cell treatment. This quantitative method is based on pulling membrane tethers from cell-derived plasma membrane spheres (PMSs) that are devoid of F-actin cortex and preserve the lipid membrane (*Figure 6A*; *Sinha et al., 2011*; *Bo and Waugh, 1989*; *Lingwood et al., 2008*). PMSs prepared from untreated siCTRL-, siCAV1-, and siPTRF-treated HUVECs were aspirated into a micropipette, allowing to control membrane tension. Membrane tethers were pulled from PMSs using optical tweezers (*Bo and Waugh, 1989*). The force, *f*, exerted on the membrane tether can be measured as a function of membrane tension ($\sigma$); it also depends on the bending rigidity ($\kappa$) following $f = 2\pi\sqrt{2\kappa\sigma}$ (*Figure 6B and C*; *Derényi et al., 2002*). We waited about 30 s after tube pulling and changing membrane tension and then checked that we reached a steady state, where lipids and membrane proteins had enough time to equilibrate (*Figure 6—figure supplement 1*). The analysis of the slope of $f^2$ versus $\sigma$ provides a measurement of the membrane bending rigidity of a PMS (*Figure 6D*). We measured a significant decrease of 30% of the bending rigidity from $\kappa_{siCTRL} = 23.4 \pm 0.9\ k_BT$ to $\kappa_{siCAV1} = 17.0 \pm 0.9\ k_BT$. Moreover, we observed no significant difference in siPTRF conditions compared to siCTRL, with a $\kappa_{siPTRF} = 23.4 \pm 1.3\ k_BT$ (*Figure 6D*). In good agreement with the theoretical predictions, these measurements establish that the expression of caveolin-1, but not of cavin-1, significantly stiffens the plasma membrane.

## Discussion

We report that siRNA-mediated knockdown of key structural and regulatory components of caveolae sensitizes endothelial cells to the formation of TEM tunnels triggered by the inhibition of RhoA-driven actomyosin contractility. Hence, inhibition of RhoA reduces by twofold the pool of caveolae invaginations at the plasma membrane. Mechanistically, we report that targeted depletion of caveolin-1 and cavin-1 caveolae decreases cell volume and increases cell spreading, which concur to decrease the

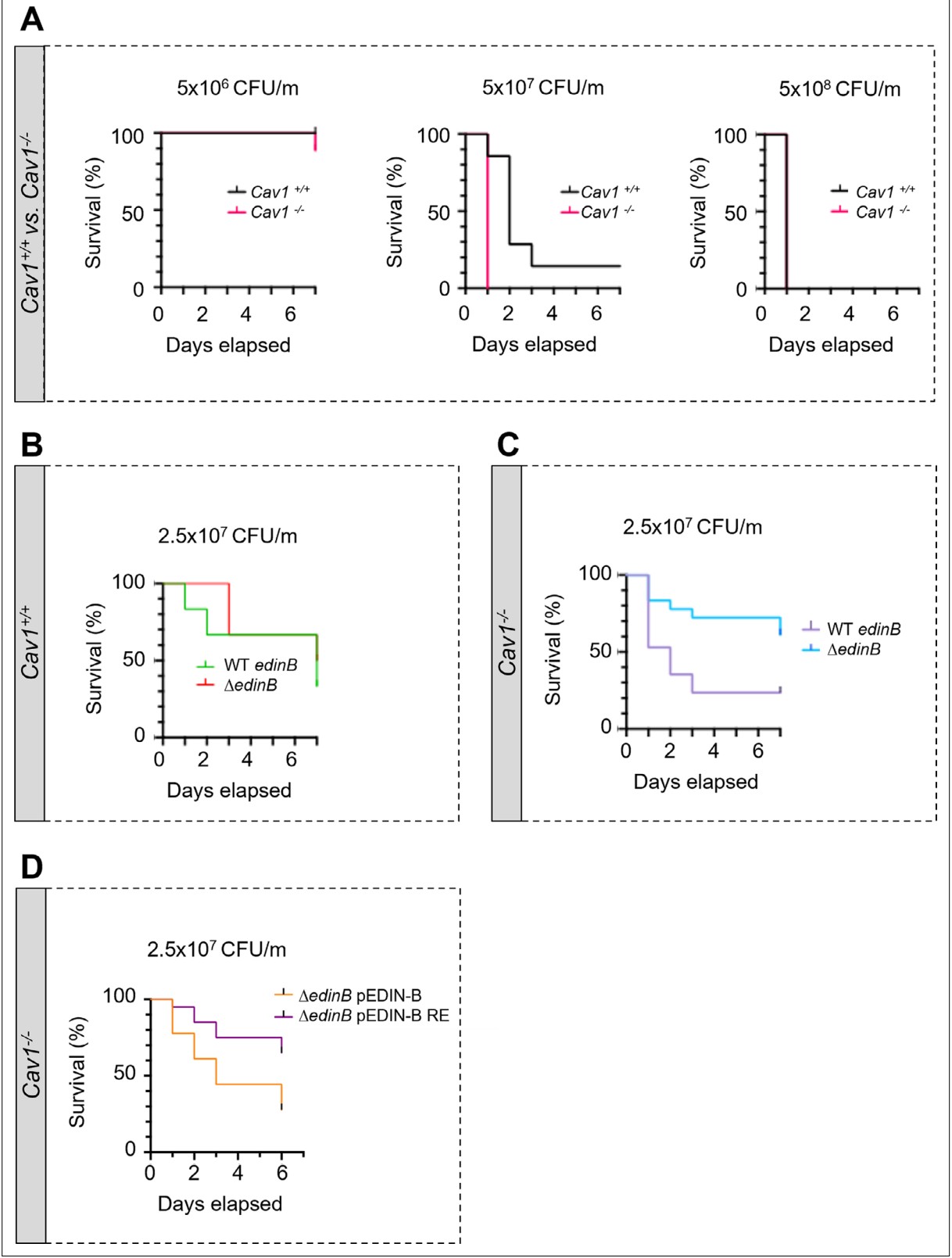

**Figure 5.** Hyper-susceptibility of Cav1-deficient mice to EDIN-B mART activity on RhoA. (**A–D**) Kaplan–Meier survival curves over 7 d for *Cav1-/-* mice and/or *Cav1+/+* littermates infected intravenously (i.v.) at day 0 with isogenic strains of *Staphylococcus aureus* and doses, expressed as colony-forming units per mouse (CFU/m). (**A**) Mice were challenged by intravenous injection of 5 × 10⁶ CFU/mouse (group of 10 *Cav1+/+* mice vs. group of 9 *Cav1-/-* mice), 5 × 10⁷ CFU/mouse (group of 7 *Cav1+/+* mice vs. group of 9 *Cav1-/-* mice) or 5 × 10⁸ CFU/mouse (group of 7 *Cav1+/+* mice vs. group of 8

*Figure 5 continued on next page*

*Figure 5 continued*

*Cav1-/-* mice). Data show a significant increase in *Cav1-/-* lethality when challenged with $5 \times 10^7$ CFU/mouse. Log-rank test (Mantel−Cox), p<0.0011 at $5 \times 10^7$ CFU/mouse (n = 1 experiment). (**B, C**) Mouse lethal doses 50 ($LD_{50}$) of WT *edinB* or Δ*edinB* strains established in *Cav1+/+* mice (**B**) and *Cav1-/-* mice (**C**). (**B**) *Cav1+/+* mice injected i.v. with $2.5 \times 10^7$ CFU/mouse (group of 12 mice for WT *edinB* and Δ*edinB* strains, n = 2 independent experiments). Log-rank test (Mantel−Cox) shows no significant difference. (**C**) *Cav1-/-* mice were injected i.v. with $2.5 \times 10^7$ CFU/mouse (groups of 17 or 18 mice for WT *edinB* or Δ*edinB* strains, n = 2 independent experiments). Log-rank test (Mantel−Cox) shows significant increase in susceptibility of *Cav1-/-* mice to WT *edinB* compared with Δ*edinB* (p=0.0123). (**D**) Comparative analysis of the susceptibility of *Cav1-/-* mice to bloodstream infection triggered by *S. aureus* Δ*edinB* complemented with a plasmid encoding wildtype EDIN-B (pEDIN-B) or the catalytically inactive EDIN-B mutant (pEDIN-B RE). Mice were injected with $2.5 \times 10^7$ CFU/mouse (groups of 20 CAV1[-/-] mice for Δ*edinB* pEDIN-B and for Δ*edinB* pEDIN-B RE strains, n = 2 independent experiments). Log-rank test (Mantel−Cox) shows a higher susceptibility of *Cav1-/-* mice infected with *S. aureus* expressing catalytically active EDIN-B (p=0.0083).

The online version of this article includes the following source data and figure supplement(s) for figure 5:

**Source data 1.** Data for *Figure 5*.

**Figure supplement 1.** *Cav1+/+* mice susceptibility to EDIN-B in staphylococcal septicemia.

**Figure supplement 1—source data 1.** Data for *Figure 5—figure supplement 1*.

height of cells favoring membrane apposition for TEM nucleation. Strikingly, we establish a specific role for caveolin-1 in controlling the opening speed of TEMs that has a dramatic impact on TEM size. In good agreement with the recorded increase in TEM formation and width following caveolin-1 knockdown, we establish a higher susceptibility of Cav1-deficient mice to lethal effects triggered by EDIN-B mART during staphylococcal bloodstream infection. Finally, we provide theoretical and experimental evidence that in addition to the well-established role of caveolae in membrane tension homeostasis, caveolin-1 rigidifies plasma membrane either directly or indirectly, as discussed below.

Our biophysical measurements unravel that caveolin-1 depletion decreases membrane bending rigidity by 30%, while depletion of cavin-1 has no impact. This experimental value is of the same order of magnitude as the 55% decrease in bending rigidity inferred from the cell dewetting model. The difference recorded between these approaches can be attributed to several approximations in the theoretical model, notably the simplified treatment of the physical membrane tension variation, the cytoskeletal contributions, and the interactions between neighboring TEM tunnels. Our tube-pulling experiments can be discussed along two lines. Indeed, since caveolin-1 is inserted in the cytosolic leaflet of the plasma membrane, when a nanotube is pulled toward the exterior of the PMS, we can expect two situations depending on the ability of caveolin-1 to deform membranes, which remains to be addressed (*Porta et al., 2022*). (1) If Cav1 does not bend membranes, it could be recruited in the nanotube at a density similar to the PMS and our force measurement would reflect the bending rigidity of the PMS membrane. Cav1 could then stiffen membrane either as a stiff inclusion at high density or/and by affecting lipid composition. (2) If Cav1 bends the membrane, it is expected from caveolae geometry that the curvature in the tube would favor Cav1 exclusion. The force would then reflect the bending rigidity of the membrane depleted of Cav1, which should be the same in both types of experiments (WT and Cav1-depleted conditions) if the lipid composition remains unchanged upon Cav1 depletion. Note that the presence of a very reduced concentration of Cav1 compared to the plasma membrane has been reported in tunneling nanotubes (TNT) connecting two neighboring cells (*Li et al., 2022*). The typical diameter of these TNTs is similar to the diameter of tubes pulled from PMS. At this stage, we cannot decipher between both properties for Cav1. Considering a direct mechanical role of Cav1, previous studies showed that inclusion of integral proteins in membranes had no impact on bending rigidity, as shown in the bacteriorhodopsin experiment (*Manneville et al., 1999*), or even decreased membrane rigidity as reported for the $Ca^{2+}$-ATPase SERCA (*Girard et al., 2005*). Previous simulations have also confirmed the softening effect of protein inclusions (*Fowler et al., 2016*). Nevertheless, our observations could be explained by a high density of stiff inclusions in the plasma membrane (>>10%), which is generally not achievable with the reconstituted membranes. Considering an impact on lipid composition, it is well established that caveolae are enriched with cholesterol, sphingomyelin, and glycosphingolipids, including gangliosides (*Parton et al., 2020b*; *Roitenberg et al., 2018*), which are known to rigidify membranes (*Rawicz et al., 2008*; *Steinkühler et al., 2019*). Thus, caveolin-1 might contribute to the enrichment of the plasma membrane with these lipid species. We did not establish experimental conditions allowing us to deplete cholesterol without compromising the shape of HUVECs, which prevented a proper analysis of TEM dynamics. Moreover, a previous attempt to increase TEMs width by softening the membrane through the incorporation of

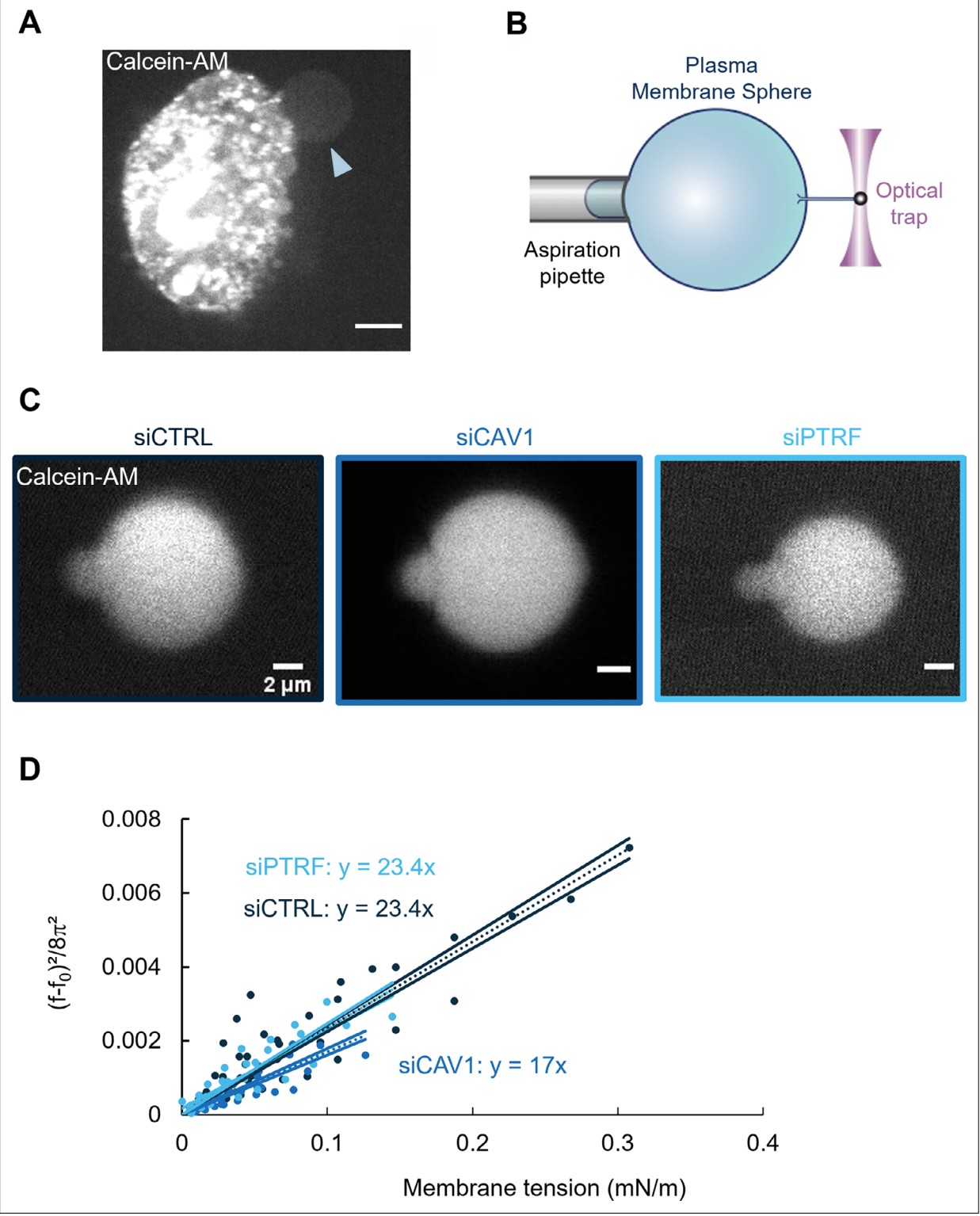

**Figure 6.** Caveolin-1 expression is a key determinant of membrane bending rigidity. (**A**) Confocal spinning disk image of a human umbilical vein endothelial cell (HUVEC) displaying a calcein-AM-positive attached plasma membrane sphere (PMS) (arrowhead). Scale bar, 10 µm. (**B**) Schematic representation of the device used for measuring membrane mechanical parameters. It shows micropipette aspiration (gray) of a PMS (blue) and a tube pulled from the PMS through a bead bound to the PMS and trapped with an optical tweezer (purple). Increasing the aspiration pressure in the pipette allowed a progressive increase in the PMS membrane tension. (**C**) Confocal images show examples of calcein-AM-positive PMSs prepared from siCTRL-, siCAV1-, or siPTRF-transfected cells during micropipette aspiration. Scale bars, 2 µm. (**D**) The force required to pull membrane tubes rescaled to ((*f*–

*Figure 6 continued on next page*

*Figure 6 continued*

$f_0)^2/8\pi^2$ (in pN$^2$), which is a function of the membrane tension (mN/m) for different siCTRL (dark blue), siCAV1 (blue), and siPTRF (light blue) treatments. The force $f_0$ was measured for each tube as the force when the membrane tension vanishes. The bending rigidity $\kappa$ (in $k_B T$) was determined via the slope of the linear regression. The data were calculated from n = 5 to n = 10 tubes per condition (>4 independent experiments). Linear regression data are shown as dashed lines, and 95% confidence intervals are shown as solid lines (slope ± SD). Data recorded between siCTRL and siCAV1 are significant showing no overlap between respective 95% confidence intervals.

The online version of this article includes the following source data and figure supplement(s) for figure 6:

**Source data 1.** Data for *Figure 6D*.

**Figure supplement 1.** Force measurements were done at equilibrium.

**Figure supplement 1—source data 1.** Data for *Figure 5—figure supplement 1*.

poly-unsaturated acyl chains into phospholipids failed, likely due to homeostatic adaptation of the membrane's mechanical properties (*Tsai et al., 2022*). Further studies are now required to establish whether and how caveolin-1 oligomers control membrane mechanical parameters through modulation of lipids organization or content. Caveolin-1 expression may also contribute to plasma membrane stiffening by interacting with membrane-associated components of the cortical cytoskeletal or structuring ordered lipid domains. Nevertheless, it has been reported that the Young's modulus of the cell cortex dramatically decreases in ExoC3-treated cells (*Ng et al., 2017*), suggesting a small additional contribution of caveolin-1 depletion to membrane softening. This is supported by 2D STORM data showing a dramatic reorganization of actin cytoskeleton in ExoC3-treated cells into a loose F-actin meshwork that is not significantly exacerbated by caveolin-1 depletion. Altogether, our results suggest that the presence of Cav1 stiffens plasma membranes, and that the exact origin of this effect must be further investigated.

Our study shows that membrane rigidity sets the maximal size of TEM aperture, although an actin ring appears before TEM closure (*Stefani et al., 2017*). Actin ring assembly and stiffening is indeed a player in TEM opening, and it is included in our differential equation describing TEM opening dynamics (*Equation 3*). In some configurations, actin ring assembly is the dominant player, such as in TEM opening after laser ablation (ex novo TEM opening), as we previously reported (*Stefani et al., 2017*). In contrast, here we investigate de novo TEM opening, for which we expect that bending rigidity can be estimated without accounting for actin assembly (*Gonzalez-Rodriguez et al., 2012*). Such a bending rigidity estimate (*Equation 5*) is obtained by considering two different time scales: the time scale of membrane tension relaxation, governed by bending rigidity, and the time scale of cable assembly, governed by actin dynamics. We expect the first time scale to be shorter, and thus the maximum size of de novo TEMs to be mainly constrained by membrane tension relaxation. However, we cannot rule out that the formation of an actin cable around the TEM before it reaches its maximum size may limit the correct estimation of the bending rigidity.

Our data establish a function to cavin-1-dependent caveolae in the regulation of cell height. Indeed, we measured an increase in cell spreading and a decrease in cell volume upon depletion of caveolin-1 or cavin-1, which are essential components of caveolae. This is consistent with findings showing that cell homeostasis during spreading is achieved by decreasing cell volume. Such a coupling between variations in cell spreading area and cell volume is breached by disrupting cortical contractility, resulting in cell swelling (*Xie et al., 2018*). Consistently, we show that the ExoC3-triggered increase in endothelial cell spreading occurs concomitantly with a decrease in cell volume. This might be a consequence of the inhibition of RhoA signaling, which is permissive for the activity of volume-regulated anion channels (*Carton et al., 2002*). Here, we show that the decrease in cell volume upon depletion of cavin-1, and to a lesser extent after depletion of caveolin-1, occurs independently of RhoA signaling, that is, in ExoC3-treated cells. Thus, how caveolae control the cellular volume once RhoA signaling is inhibited awaits determination. Our findings also suggest that depletion of caveolae and RhoA inactivation may release membrane-folding-driven mechanical constraints that would limit cell spreading. Another hypothesis is that caveolae control indirectly the cellular geometry via a documented function of cavin-1 on cell signaling (*Derényi et al., 2002*; *Li et al., 2022*; *Manneville et al., 1999*; *Girard et al., 2005*; *Fowler et al., 2016*; *Parton et al., 2020b*; *Roitenberg et al., 2018*; *Rawicz et al., 2008*; *Xie et al., 2018*; *Carton et al., 2002*; *Jansa et al., 2001*; *Jansa and Grummt, 1999*; *Liu and Pilch, 2016*; *Shi et al., 2015*).

Our study points to the underlying mechanisms by which caveolae regulate the frequency of TEM nucleation. Nucleation of TEMs requires the apposition of the basal and apical cell membranes, which is hindered by the intermembrane distance, set by the cell height. Meeting of the two membranes may create an initial precursor tunnel, which needs to be sufficiently big to enlarge into an observable TEM, instead of simply closing back. The size of the minimal precursor tunnel required to give rise to a TEM increases with membrane bending rigidity and decreases with membrane tension (*Gonzalez-Rodriguez et al., 2012*). Silencing cavin-1 or caveolin-1 both lead to a decrease in cell height, thus favoring the likelihood of precursor tunnel nucleation. While silencing cavin-1 has no significant impact on either membrane tension or bending rigidity, silencing caveolin results in both an increase in membrane tension and a decrease in bending rigidity, which results in a decrease in the required minimal radius of the precursor tunnel, thus further favoring TEM nucleation. Overall, our results offer a consistent picture of the physical mechanisms by which caveolae modulate TEM nucleation.

The molecular basis of interindividual variability in *S. aureus* infections is unclear, although the expression of caveolin-1 has recently been suggested (*Spaan et al., 2022*). Indeed, a human genetics study pinpointed the impact of haploinsufficiency of the OTULIN deubiquitinase in sensitizing host cells to lysis triggered by the highly prevalent α-hemolysin pore-forming toxin of *S. aureus*, a major determinant in staphylococcal pneumonia (*Spaan et al., 2022*). The defect in OTULIN activity led to an increase in caveolin-1 cellular protein levels in dermal cells, which in turn upregulates the expression of the ADAM10 receptor of α-hemolysin pore-forming toxin. Here, we report that caveolin-1 expression is not regulated by RhoA signaling. Furthermore, we establish that the loss of caveolin-1 expression sensitizes mice to the lethal effects triggered by EDIN-B mART activity on RhoA during *S. aureus* septicemia, thus ascribing a pathogenicity potential to this exotoxin that is counteracted by the expression of caveolin-1. In line with this, a recent analysis of an observational cohort of patients with *S. aureus* community-acquired pneumonia indicated that EDIN-B, together with Panton–Valentin leukocidin, are positively associated with the etiology of hemoptysis, manifested by the presence of blood in sputum (*Gillet et al., 2021*). Whether TEMs are involved in the etiology of hemoptysis remains to be investigated. Although we cannot argue that there is a causal link between the lethal effect of EDIN-B in caveolin-1-deficient animals and the protective function of caveolin-1 against TEM tunnel formation in endothelial cells, these findings represent a first step toward establishing that endothelial cells regulate TEM tunnel formation and width *in vivo*. Such a hypothesis is supported by previous findings showing that RhoA inhibition reduces paracellular permeability while dramatically increasing transcellular permeability, and that ADP-ribosyltransferases targeting RhoA increase vascular permeability to the point of animal death when TEM enlargement is no longer controlled (*Boyer et al., 2006*; *Rolando et al., 2009*). Collectively, these findings ascribe to variations in caveolin-1 expression a bidirectional impact on host susceptibility to *S. aureus* infection, pointing to a possible role of TEM tunnels.

## Materials and methods

### Bacterial strains and culture conditions

*S. aureus* HT20020209-LUG1799, referred to as wildtype WT *edinB* in this study, is a minimally passaged ST80 *SCCmecIV* PVL+ MRSA strain isolated in France (deposited at http://www.ebi.ac.uk/ena) that is representative of ST80 CA-MRSA clones (*Perret et al., 2012*; *Tristan et al., 2007*). *S. aureus* HT20020209-LUG1799 with the *edinB* gene deletion is referred to as Δ*edinB* in this study (*Courjon et al., 2015*). The strain Δ*edinB* was complemented with a pMK4-pPROT plasmid expressing the wildtype form of EDIN-B to generate the strain Δ*edinB* pEDIN-B or with pMK4-pPROT expressing the catalytically inactive mutant form of EDIN-B (EDIN-B-R185E) to generate the strain Δ*edinB* pEDIN-B RE, as previously described (*Courjon et al., 2015*). Both strains were grown in lysogeny broth (LB) with shaking at 200 rpm and 37°C. Equal growth kinetics were verified (not shown).

### Mouse infection model

Adult male and female B6.Cg-*Cav1^{tm1Mls}*/J mice (strain # 007083, Jackson Laboratory) and C57BL/6J mice (Charles River) were housed under specific pathogen-free conditions at the Institut Pasteur animal facilities licensed by the French Ministry of Agriculture (B75150102). Mice received food and water ad libitum, and their weight was recorded daily throughout the study. *S. aureus* strains were

cultured in LB at 37°C until reaching an $OD_{600}$ = 1 after overnight culture. After washing twice in PBS, cell pellets were resuspended in sterile 0.9% NaCl. Infections were carried out by injecting 300 µl serial dilutions of inoculum intravenously into the tail vein of the mice. Animal survival was monitored daily. The size of animal groups was calculated based on previous data and a power analysis assuming a risk of 5% and a power of 80%.

## Cell culture, transfection, and toxicity

Primary HUVECs (PromoCell) were cultured in human endothelial serum-free medium (SFM, Gibco) containing 20% fetal bovine serum, 20 ng/ml FGF-2, 10 ng/ml EGF, and 1 µg/ml heparin and referred to as complete SFM (SFMc). Cells were cultured on gelatine-coated culture ware at 37°C with 5% $CO_2$ for as many as six passages. For siRNA transfection, HUVECs were cultured at a density of 38,000 cells/ $cm^2$. ON-TARGETplus smart pool siRNA (Dharmacon) targeting human caveolin-1 (L-003467-00-0005), EHD2 (L-016660-00-0005), cavin-1/PTRF (L-012807-02-0005) or control RNAi SR-CL000 (Eurogentec) was used at 100 nM via magnetofection technology (OZ Biosciences) following the manufacturer's instructions in serum-free OptiMEM (Gibco). When necessary, cells were electroporated 24 hr post-magnetofection and then used from 48 to 54 hr post-magnetofection. HUVEC electroporation of plasmids encoding caveolin-1-GFP (*Sinha et al., 2011*) or LifeAct-GFP (ibidi, GmbH, Planegg/Martinsried, Germany) was performed as described in *Doye et al., 2002*. Briefly, cells were trypsinized and resuspended in Ingenio Solution (Mirus) containing plasmid DNA (10 µg/$10^6$ cells) in a 4 mm cuvette (CellProjects). Cells were then electroporated at 300 V and 450 µF with one pulse of a GenePulser electroporator (Bio-Rad). Cells were incubated in SFMc, and the medium was replaced 3 hr post-electroporation. Cells were treated 6 hr post-electroporation. Recombinant ExoC3 and EDIN were produced in *Escherichia coli* and purified as described in *Boyer et al., 2006*. Cells were treated with ExoC3 or EDIN exotoxins at a final concentration of 100 µM for 24 hr.

## Immunoblotting and western blotting

Proteins were resolved on 12% SDS-polyacrylamide gels and transferred to nitrocellulose membranes (GE Healthcare). The primary antibodies used were mouse anti-EHD2 (L-05) (Santa Cruz sc-100724), rabbit anti-Cav1 (Cell Signaling Technology #3238), rabbit anti-PTRF (Proteintech, 18892-1-AP), and mouse anti-GAPDH (Santa Cruz sc-47724). The secondary antibodies used were HRP-conjugated anti-mouse or anti-rabbit (Dako). Signals were imaged using an Imager-680 system from Amersham and were quantified with ImageJ software.

## Immunofluorescence

HUVECs were seeded on a gelatine-coated µ-Dish 35 mm, high (ibidi), and treated as indicated previously (*Stefani et al., 2017*). Cells were fixed in ready-to-use paraformaldehyde (PFA), 4% in PBS (Bio-Rad). Immunostaining of fixed cells permeabilized in 0.5% Triton X-100 was performed. FITC-phalloidin or TRITC-phalloidin at 1 µg/ml (Sigma) were used to stain actin, and DAPI (Life Technologies) was used to label nuclei.

## Video microscopy

HUVECs were electroporated with LifeAct-GFP-pCMV as described above and seeded on a gelatine-coated polymer coverslip dish (ibidi). After treatment with ExoC3 (see above), the cells were supplemented with 25 mM HEPES (pH 7.4) and their proliferation was recorded at 37°C with a Nikon $T_i$ inverted microscope using an Ultraview spinning disk confocal system (PerkinElmer). For determining TEM opening, images were taken every 10–25 s for 1 hr. To determine the opening speed ($S_o$), images were taken every second for 30 min. Acquired videos were analyzed via an ICY-based semi-automatic protocol.

## Cell volume measurement

After siRNA treatment and other treatments, cells were seeded onto a PDMS chip ($2.10^6$ cells/ml) as described previously (*Zlotek-Zlotkiewicz et al., 2015*). Briefly, chambers were coated with 10 µg/ml fibronectin in PBS (Life Technologies) for 1 hr at room temperature (RT). Chambers were washed with medium before cell seeding. Cells were resuspended in medium supplemented with 0.5 mg/ml Alexa Fluor 488 dextran (MW 10 kD; Life Technologies) and then injected into the chamber. Finally,

the chamber was immersed in medium to prevent evaporation. HUVECs were allowed to adhere for 4–6 hr in SFMc at 37°C with 5% $CO_2$ before acquisition.

Images were analyzed with MATLAB (MathWorks). The intensity of the background was maximal in the absence of any object ($I_{max}$) and represented the fluorescence value for the maximal height of the chamber ($h_{max}$). In contrast, the pillar excludes fluorescence, which therefore reflects the minimal fluorescence intensity ($I_{min}$). At the cell level, the fluorescence was partially excluded at a given point for a cell at a given height ($h_{x,y}$). This strategy enables the measurement of the fluorescence intensity at this point ($I_{x,y}$). After calibrating the fluorescence intensity signal $\alpha = (I_{max} - I_{min})/h_{max}$, integration of the fluorescence intensity over the cell area provided the cell volume ($V_{cell}$).

## Metal replicates and transmission electron microscopy

Metal replicates of the ventral plasma membranes of HUVECs cultured on glass coverslips were obtained by sonication according to a published protocol (*Lemerle et al., 2023*; *Heuser, 2000*). Briefly, cells were rinsed three times with Ringer's buffer with $Ca^{2+}$ and then briefly subjected to a concentration of 0.5 mg/ml poly-L-lysine diluted in $Ca^{2+}$-free Ringer's buffer (Sigma-Aldrich). Poly-L-lysine was removed by washing with $Ca^{2+}$-free Ringer's buffer. The coverslips were immersed in KHMgE buffer at 37°C before sonication (Vibra-Cell VCX130 ultrasonic processor, Sonics) at a 20% amplitude. The unroofed cell membranes were then immediately fixed for 30 min with 2% glutaraldehyde/2% PFA. The cell membranes were sequentially treated with 1% $OsO_4$, 1.5% tannic acid, and 1% uranyl acetate before dehydration via successive ethanol baths, which was ultimately substituted with hexamethyldisilazane (#C16700-250; LFG Distribution).

For immunogold labeling, sonicated plasma membranes were fixed with 4% PFA before incubation with primary (GFP Thermo Fisher A11122 rabbit 1/20) and secondary antibodies coupled to gold beads (rabbit-gold 815.011 AURION goat 1/25), and the membranes were then incubated with a $NaBH_4$ solution to inactivate aldehydes. The membranes were finally fixed with 2% glutaraldehyde and subjected to the same treatment as that used for morphology studies. The dehydrated samples were metalized via rotary metallization. The coverslips were placed in the chamber of a metallizer (ACE600, Leica Microsystems). Once under a high vacuum (>$10^{-5}$ mBar), the membranes were covered with 2 nm of platinum stabilized by 4–6 nm of braided carbon. The resulting platinum replicas were separated from the glass by flotation on acid, washed several times in distilled water baths containing 0.1% detergent (one drop in 10 ml, Photo-Flo, Kodak), and placed on electron microscopy (EM) grids covered with a carbon film (200 mesh formvar/carbon, LFG Distribution). The grids were mounted in the goniometer with eucentric side entry of a transmission electron microscope operating at 80 kV (CM120, Philips), and images were recorded with a Morada digital camera (Olympus). The images were processed with ImageJ software to adjust brightness and contrast and are presented in reverse contrast.

## RhoA mono-ADP-ribosylation assay

HUVECs were seeded at a density of 27,000 cells/$cm^2$ transfected with siCTRL, siCAV1, and siPTRF and then left untreated or treated with ExoC3. Cells were lysed in ADP-ribosylation buffer (20 mM Tris–HCl, 1 mM EDTA, 1 mM DTT, 5 mM $MgCl_2$ and cOmplete protease inhibitor EDTA-free [Roche], pH 7.5) and passed through a 27 G syringe 20 times. Cell lysates were collected by centrifugation at 12,000 × $g$ for 10 min, and the protein concentration was determined by BCA assay (Thermo Fisher Scientific). The reaction was carried out by incubating 20 µg of cell lysate with 2 µg of ExoC3 and 10 µM 6-biotin-17-$NAD^+$ (BioLog) at 37°C for 30 min. The reaction was terminated by the addition of 1 mM DTT and Laemmli buffer (0.3 M Tris–HCl, 10% SDS, 37.5% glycerol, and 0.4 mM bromophenol blue) and boiling at 100°C for 5 min. The samples were subjected to 12% SDS–PAGE, and the proportion of ADP-ribosylated (i.e., biotin-ADPr-RhoA) RhoA in the sample was measured by western blotting using streptavidin–peroxidase.

## Plasma membrane sphere formation and tether extraction

PMS were generated via a protocol adapted from *Lingwood et al., 2008*. Cells were grown to 60–80% confluence on gelatine-coated 100 mm dishes and incubated for 6–8 hr in PMS buffer (1.5 mM $CaCl_2$, 1.5 mM $MgCl_2$, 5 mM HEPES, 2 mg/ml glucose, 150 mM NaCl, 10 µM MG132 in PBS [pH 7.4]) to induce membrane swelling of the PMSs. Individual PMSs were aspirated using a casein-passivated

micropipette connected to a piezo-stage (PI, Karlsruhe, Germany) for manipulation and to a hydro-static aspiration control system (*Cuvelier et al., 2005*). Micropipettes were made in-house with boro-silicate capillaries pulled into fine cones using a laser pipette puller (P-2000, Sutter Instrument Co.) and microforged at the desired inner diameter (3–4 μm), as described previously (*Sinha et al., 2011*). For the extraction of tethers from PMSs, we used optical tweezers built in-house that consisted of a single fixed laser beam (infrared laser wavelength of 1070 nm) focused through a ×60 water objective mounted on a confocal microscope (Nikon TE2000 inverted microscope) (*Sorre et al., 2012*). To pull tethers from PMSs, we coated streptavidin beads (3 μm in diameter, Spherotech) with fibronectin, which allowed the beads to bind to PMSs. A membrane tether was generated by bringing the micropipette-held PMS into contact with a bead trapped by the optical tweezers, and then by moving the PMS away from the bead. After extraction, the tether was held at a constant length between 2 and 5 μm, and tether forces were measured during gradual increase in aspiration pressure and thus PMS tension. At a given membrane tension, the corresponding tether force was measured at least 30 s after the pressure change and when equilibration was established (see representative force vs. time curve in *Figure 6—figure supplement 1*). Analysis was performed as described below. Briefly, for each membrane tether, the tether force ($f$) was plotted as a function of the square root of the membrane tension ($\sqrt{\sigma}$) calculated using the Laplace law (*Evans and Rawicz, 1990*) and corrected when the length of the 'tongue' of a PMS tether inside the micropipette was shorter than the radius of the pipette (*Guevorkian et al., 2021*). For each tether, we determined a force $f_0$ corresponding to the intercept of the linear regression ($f$ vs. $\sqrt{\sigma}$). To estimate the bending rigidity for PMSs obtained from membranes of cell treated differently (siCRTL, siCAV1, and siPTRF cells), the data obtained for all tethers were pooled and rescaled as $(f-f_0)^2/8\pi^2$ and plotted as a function of membrane tension. The corresponding PMS rigidity was obtained from the slope of the linear fit calculated using GraphPad Prism.

## Single-molecule localization microscopy of F-actin

For single-molecule localization microscopy (SMLM) of F-actin, we used 2D STORM. HUVECs were cultured on 1.5 high-performance coverslips coated with 10 μg/ml fibronectin for 2 hr at RT. Soluble G-actin was pre-extracted (0.25% Triton, 0.1% glutaraldehyde in 80 mM PIPES, 5 mM EGTA, and 2 mM MgCl$_2$, pH 6.8) for 30 s at 37°C. The cells were then treated with glutaraldehyde (0.25% Triton, 0.5% glutaraldehyde in 80 mM PIPES, 5 mM EGTA, 2 mM MgCl$_2$, pH 6.8) for 10 min at 37°C. Glutaraldehyde was quenched for 7 min at RT (0.1% NaBH$_4$ in PBS). Saturation was reached after 1 hr at RT (0.22% gelatine and 0.1% Triton X-100 in PBS, pH 7.3). Actin filaments were stained overnight at 4°C with phalloidin-AF647 (500 nM in PBS). Cells were then placed in switching buffer freshly prepared before imaging (50 mM Tris, 10 mM NaCl, 10% glucose, 0.5 mg/ml glucose oxidase, 50 μg/ml catalase, and 50–100 mM cysteamine [MEA, pH 8.3]) in the presence of phalloidin A647 (10–50 nM). Super-resolved images were acquired with an Elyra-7 SMLM microscope (Carl Zeiss, Germany) using an N.A = 1.4 oil ×63 objective (Carl Zeiss) with 1.518 refractive index oil (Carl Zeiss) and a pco.edge 4.2 camera system (Excelitas PCO GmbH). All processing analyses were performed with Zen software (Carl Zeiss). Localization of individual molecules was determined using a peak mask size of 15-pixel diameter and a peak intensity-to-noise ratio between 5 and 6. Drift was corrected using a model-based algorithm from Zen software. Super-resolution images were reconstructed at a 20 nm pixel size.

## Mesh analysis of the F-actin network imaged via 2D STORM

To quantify actin mesh size from the super-resolution images, we developed an image analysis workflow to reliably segment the delimiting continuous actin filaments. First, we clipped the images to a maximum detection count of 10 and applied a line enhancement filter to suppress short apparent discontinuities by modifying a previously described approach (*Al Jord et al., 2022*). Specifically, the maxima of the mean intensities along line profiles at 15 equally spaced orientations around each pixel were calculated (line length 21 px or approximately 0.4 μm). Subsequently, tube-like structures were extracted from the line-enhanced images by applying a Hessian filter implemented in scikit-image (smoothing width of 2 px) as described previously (*van der Walt et al., 2014*) with results binarized using Otsu thresholding. The obtained masks were subdivided into putative filament segments by defining sections of equal length along the mask's morphological skeleton and assigning each pixel to its nearest segment (the chosen length was 20 px or approximately 0.4 μm). The final actin filament

segmentation was performed considering only segments that (1) were not morphologically connected to at least one other segment, to exclude non-filament detections, or (2) exhibited a mean detection count lower than the super-resolution image Otsu threshold. Finally, the actin meshwork was obtained as the connected components of the inverted actin filament segmentation mask.

## Physical modeling

TEM dynamics were theoretically interpreted based on the generalization of our earlier model, which was used for a single TEM (*Gonzalez-Rodriguez et al., 2012*; *Stefani et al., 2017*), to account for the case when several TEMs open simultaneously. This generalization was previously used to interpret the data in *Tsai et al., 2022*, and we present it here for completeness. The opening of a TEM is driven by a net force,

$$F_d = 2\sigma - \frac{T}{R},$$  (1)

where σ is the membrane tension, $T$ is the line tension, and $R$ is the TEM radius. For a model lipid membrane in the entropic regime, membrane tension $\sigma$ depends on $R$, quantified by the Helfrich's law. Written in a generalized form to describe $N$ simultaneous TEMs in the same cell, Helfrich's law states that

$$\sigma = \sigma_0 \exp\left[-\frac{\sum\limits_{i=1}^{N} R_i^2}{R_c^2}\right]$$  (2)

where $R_c^2 = \left(R_{\text{cell}}^2 k_B \hat{T}\right) / (8\pi\kappa)$, $R_{\text{cell}}$ is the radius of the cell, $k_B$ is the Boltzmann constant, $\hat{T}$ is the temperature, and $\kappa$ is the effective bending rigidity of the cell membrane, which quantifies the energy required to bend the membrane. *Equation 2* treats the effect of several simultaneous TEMs in an additive manner. This approximation is used here to predict TEM size because at maximum opening of simultaneous TEMs their respective membrane relaxation is felt by each other as it can be inferred from the shape that neighboring TEMs adopt in experiments. This additive treatment would appear less appropriate to describe the likelihood of nucleating a second TEM in the presence of a first one (a calculation that is not performed here) since membrane relaxation by a TEM may not be felt at membrane regions distant from it. While rigorously derived for a pure lipid membrane, we assumed that Helfrich's law is applicable to describe the relationship between the effective membrane tension $\sigma$ acting on TEMs and the observed projected surface in our cells. We expect Helfrich's law to be applicable on short time scales, before active cell tension regulation takes place (*Sens and Plastino, 2015*), especially in cases where cytoskeletal forces play a modest role, such as for red blood cells (*Helfrich, 1973*) or for the highly disrupted cytoskeletal structure of our intoxicated cells. Thus, the parameter $\kappa$ in *Equation 2* is an *effective* bending rigidity, whose value may somewhat differ from that of a pure lipid membrane to account for the role played by protein inclusions and the mechanical contribution of the remaining cytoskeletal elements after cell treatment with the toxin. Moreover, we assumed that an eventual buffering role played by the remaining caveolae can be described through modification of the effective parameters $\sigma_0$ and $\kappa$. A limitation of our theoretical description arises from the use of spatially uniform changes in parameter values to describe the differences between experimental conditions, thus assuming spatially uniform effects. However, we cannot exclude the existence of nonuniform effects, such as changes in the size and organization of the remaining actin mesh, which could set local, nonuniform barriers to TEM enlargement in a manner not accounted for by our model.

As discussed by *Gonzalez-Rodriguez et al., 2012*, the effective line tension is not a constant; it increases with time due to the formation of an actomyosin cable around the TEM. Stefani et al. described this increase as linear, $T \sim \alpha t$. The dynamics of TEM opening are derived from a balance between the driving force (*Equation 2*) and cell–substrate friction. Assuming $N$ identical TEMs, this balance is

**Table 2.** Estimate of the variation of mechanical cell parameters between different experimental conditions.

The value of the effective membrane tension $\sigma_0$ for the control case is obtained from earlier estimates (**Doye et al., 2002**). The increase in $\sigma_0$ for siCAV1 and siPTRF cell membranes is deduced from our experimental data using **Equation 9**. The transendothelial cell macroaperture (TEM) maximum area, $A_{max}$, is the median not the average value because the median is a more robust estimator in the presence of a few extremely large values. As discussed in the text, the variations in the bending rigidity are roughly proportional to variations in $(N \cdot A_{max})^{-1}$, where $N$ is the average number of TEMs opening simultaneously and $A_{max}$ is the TEM maximum area.

| Condition | $\sigma_0$ (µN/m) | $A_{max}$ (µm²) | $N$ (average) | $(N \cdot A_{max})^{-1}$ (µm⁻²) |
|---|---|---|---|---|
| Control | 25 ± 10 | 8.1 ± 0.5 | 0.90 ± 0.09 | 0.137 ± 0.022 |
| siCAV1 | 50 ± 7 | 23 ± 4 | 1.92 ± 0.15 | 0.023 ± 0.006 |
| siPTRF | 29 ± 7 | 9.3 ± 0.7 | 1.34 ± 0.11 | 0.080 ± 0.012 |

$$2\sigma_0 R \exp\left(-\frac{NR^2}{R_c^2}\right) - \alpha t = \mu R^2 \frac{dR}{dt}, \tag{3}$$

where $\mu$ is the friction coefficient. While this equation can be solved numerically, the following analytical approximation can also be obtained, as derived in **Tsai et al., 2022**:

$$x \exp(x^2) \approx \frac{8\sigma_0^2 \sqrt{N}}{\mu \alpha R_c}, \tag{4}$$

where $x = N^{1/2} R_{max}/R_c$ and $R_{max}$ is the maximum TEM radius. As discussed in **Tsai et al., 2022**, for biologically relevant values of the model parameters, **Equation 4** leads to the following approximate estimate of the effective membrane bending rigidity:

$$\kappa \approx \frac{k_B \hat{T}}{8\pi} \frac{A_{cell}}{N A_{max}}. \tag{5}$$

Considering the experimental results shown in **Table 2** and using the estimate $\kappa \sim (N \cdot A_{max})^{-1}$ based on **Equation 5**, we deduced an effective bending rigidity decrease after silencing caveolin-1. We note that the estimate of $\kappa$ provided by **Equation 5** is independent of $\alpha$ and thus of actin cable assembly. This simplification arises from membrane tension relaxing over a shorter time scale than actin assembly. Thus, we expect the maximum size of de novo TEMs to be mainly constrained by membrane tension relaxation (**Gonzalez-Rodriguez et al., 2012**), unlike ex novo TEM enlargement upon laser ablation, for which the dynamics of actin cable assembly control TEM opening (**Stefani et al., 2017**). The approximate estimates given in **Table 2** can be refined using **Equation 4** directly, which also considers variations in membrane tension. The following expression for the bending rigidity is thus obtained:

$$\kappa \approx \frac{k_B \hat{T}}{8\pi} \frac{R_{cell}^2}{N R_{max}^2} \ln\left(\frac{8\sigma_0^2}{\mu \alpha R_{max}}\right). \tag{6}$$

**Equation 6** is used to obtain the predictions of bending rigidity variations mentioned in the text: a reduction in effective bending rigidity by a factor of approximately 1.2 for siPTRF cell membranes and by a factor of approximately 2.3 for siCAV1 cell membranes, which is equivalent to a 15% decrease for siPTRF and 55% for siCAV1 cell membranes.

## Fit of the initial opening speed

For short time intervals, the differential equation for the opening dynamics can be simplified to

$$\mu R^2 \frac{dR}{dt} = 2\sigma_0 \left(1 - \frac{R_n}{R}\right), \tag{7}$$

where $R_n = T/(2\sigma_0)$ is the minimal nucleation radius. This equation can be integrated to obtain

$$R_n^2 \ln\left(\frac{R}{R_n} - 1\right) + R_n R + \frac{R^2}{2} - C = \frac{2\sigma_0}{\mu} t, \tag{8}$$

where $C$ is an integration constant, whose value is such that $R = R_0$ for $t = 0$, with $R_0$ the unknown nucleated TEM radius, which is larger than the minimal nucleation radius $R_n$; $R_0 > R_n$. Because all TEM measurements reflect $R \gg R_0 > R_n$, the dominant term on the left-hand side is the term proportional to $R^2 = A/\pi$, where $A$ is the TEM area. These considerations yield the following estimate of $\sigma_0$:

$$\frac{\sigma_0}{\mu} = \frac{A_2 - A_1}{4\pi \Delta t} \tag{9}$$

where $\Delta t = 1$ s is the time interval between two acquisitions; $A_1$ is the TEM area in the first image after TEM opening (taken on average at a time $\Delta t/2$), and $A_2$ is the TEM area in the second image.

## Statistical analysis

Statistical tests were performed using the R Software v4.2.1 (R Foundation for Statistical Computing, Vienna, Austria; https://www.R-project.org/), along with packages lme4 (1.1–30) (*Bates et al., 2015*), lmerTest (v3.1–3) (*Kuznetsova et al., 2017*), emmeans (v1.8.0), except for the F-actin mesh size where the Prism Software was used. All experiments were repeated at least three times to ensure reproducibility and corresponding variability was accounted for in statistical analyses using mixed-effect regressions, with a random intercept estimated for every technical replicate. In particular, binomial outcomes (e.g., presence/absence) were analyzed using mixed-effect logistic regression. Continuous measurements (e.g., TEM area) were investigated with a Gamma log-link function in a mixed-effect framework. Contrasts between multiple pairs of categorical predictors were calculated through estimated marginal means with Tukey's correction. p-Values were calculated as indicated in the respective figure legends. p-Values were considered statistically significant at p≤0.05. Significance levels are indicated as follows: ns: not significant if p>0.05; *p≤0.05; **p≤0.01; ***:p≤0.001; ****p≤0.0001.

## Acknowledgements

C Morel and E Lemerle were supported by PhD fellowships from the University Paris Cité Doctoral School BioSPC and Sorbonne University Doctoral School complexité du vivant, respectively. We thank Marie Anne Nahori, Amel Mettouchi, Arnaud Echard, Virginia Ribeiro de Andrade (Institut Pasteur, Paris, France), and Michael Henderson (Institut Pasteur/Institut Curie, Paris France) for technical advice, sharing reagents, and discussions. We thank Pierre Nassoy (LP2N, Bordeaux, France), Francoise Brochard Wyart (Institut Curie, Paris, France), and Anne Blangy (CRBM, CNRS, Montpellier, France) for insightful discussions. We thank Darius Koster for sharing results and fruitful discussions on tether pulling experiments of plasma membrane spheres. We thank Gaëlle Letort (Department of Developmental and Stem Cell Biology, Institut Pasteur, CNRS UMR 3738) for helpful advice on image analysis. We also thank the UTechS PBI, which is part of the France–BioImaging infrastructure network (FBI) supported by the French National Research Agency (ANR-10-INBS-04; Investments for the Future) and acknowledge support from Institut Pasteur, ANR/FBI, the Région Ile-de-France (program DIM1HEALTH), and the French Government Investissement d'Avenir Programme-Laboratoire d'Excellence 'Integrative Biology of Emerging Infectious Diseases' (ANR-10-LABX-62-IBEID) for the use of the Zeiss ELYRA7 microscope.

## Additional information

### Competing interests

Patricia Bassereau: Reviewing editor, *eLife*. The other authors declare that no competing interests exist.

## Funding

| Funder | Grant reference number | Author |
|---|---|---|
| Agence Nationale de la Recherche | ANR-10-LABX-62-IBEID | Emmanuel Lemichez |
| Institut Pasteur | | Emmanuel Lemichez |

The funders had no role in study design, data collection and interpretation, or the decision to submit the work for publication.

## Author contributions

Camille Morel, Conceptualization, Data curation, Formal analysis, Investigation, Methodology, Writing – original draft, Writing – review and editing; Eline Lemerle, Investigation, Visualization, Methodology; Feng-Ching Tsai, Conceptualization, Data curation, Formal analysis, Investigation, Methodology, Writing – review and editing; Thomas Obadia, Software, Investigation, Methodology, Writing – review and editing, Statistical analysis; Nishit Srivastava, Maud Marechal, Investigation, Methodology; Audrey Salles, Methodology; Marvin Albert, Formal analysis, Methodology; Caroline Stefani, Investigation, Methodology, Writing – review and editing; Yvonne Benito, François Vandenesch, Resources; Christophe Lamaze, Validation, Writing – review and editing; Stéphane Vassilopoulos, Matthieu Piel, Supervision, Methodology; Patricia Bassereau, Conceptualization, Data curation, Formal analysis, Supervision, Methodology, Writing – original draft, Writing – review and editing; David Gonzalez-Rodriguez, Conceptualization, Supervision, Methodology, Writing – original draft, Writing – review and editing, Theoretical modeling; Cecile Leduc, Conceptualization, Formal analysis, Supervision, Investigation, Methodology, Writing – original draft, Writing – review and editing; Emmanuel Lemichez, Conceptualization, Resources, Data curation, Formal analysis, Supervision, Funding acquisition, Validation, Investigation, Visualization, Methodology, Writing – original draft, Project administration, Writing – review and editing

## Author ORCIDs

Feng-Ching Tsai 
Nishit Srivastava 
Caroline Stefani 
François Vandenesch 
Stéphane Vassilopoulos 
Patricia Bassereau 
David Gonzalez-Rodriguez 
Cecile Leduc 
Emmanuel Lemichez 

## Ethics

Adult male and female B6.Cg-Cav1tm1Mls/J mice (strain #: 007083, Jackson Laboratories) and C57BL/6J mice (Charles River) were housed under specific-pathogen-free conditions at the Institut Pasteur animal facilities licensed by the French Ministry of Agriculture (B75150102).

Reviewer #2 (Public Review): https://doi.org/10.7554/eLife.92078.3.sa1
Author Response https://doi.org/10.7554/eLife.92078.3.sa2

# Additional files

## Supplementary files

- MDAR checklist

## Data availability

All data generated or analysed during this study are included in the manuscript and supporting files.

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
