## [Editor Report · eLife assessment]

This **important** study identifies the role of caveolin-1 and cavin1 as regulators of transendothelial macroaperture (TEM). The methodology used is rigorous and **compelling**, and further research can point to more mechanistic understanding of the process.

---

## [Referee Report · Reviewer #2 (Public Review)]

Summary:

The manuscript by Morel et al. aims at identifying some potential mechano-regulators of transendothelial cell macro-aperture (TEM). Guided by the recognized role of caveolar invaginations in buffering the membrane tension of cells, the authors focused on caveolin-1 and associated regulator PTRF. They report a comprehensive *in vitro* work based on siRNA knockdown and optical imaging approach complemented with an *in vivo* work on mice, a biophysical assay allowing to measure the mechanical properties of membranes and a theoretical analysis inspired from soft matter physics.

The authors should be complimented for this multi-facetted and rigorous work. The accumulation of pieces of evidence collected from each type of approach makes very convincing the conclusion drawn by the authors on the new role of cavolin-1 as an individual protein instead of the main molecular component of caveolae. On a personal note, I was very impressed by the quality of STORM images (Fig. 2) which are very illuminating and useful, in particular for validating some hypotheses of the theoretical analysis.

While this work pins down the key role of caveolin-1, its mechanism remains to be further investigated. The hypotheses proposed by the authors in the discussions about the link between caveolin and lipids/cholesterol are very plausible though challenging. Even though we may feel slightly frustrated by the absence of data in this direction, the quality and merit of this paper remain.

The analogy with dewetting processes drawn to derive the theoretical model is very attractive. However, and although part of the model has already been published several times by the same group of authors, the validity of the Helfrich formalism is a key assumption that has to be explained clearly. Here, for the first time, thanks to these STORM analysis, the authors show that HUVECs intoxicated by ExoC3 exhibit a loose and defective cortex with a significantly increase mesh size, which supports this hypothesis.

---

## [Author Response]

The following is the authors’ response to the original reviews.

We thank the Editors and the Reviewers for their comments on the importance of our work “showing a new role of caveolin-1 as an individual protein instead of the main molecular component of caveolae” in contributing to membrane bending rigidity and for constructive and thoughtful remarks that have allowed us to improve the manuscript.

Indeed, we here establish the contributing role of caveolin-1 to membrane mechanics by a molecular mechanism that needs to be further addressed. To that respect, we thank the reviewers for suggesting avenues to improve the presentation and discussion of our hypotheses based on results of theoretical model and independent biophysical measurements of membrane mechanics in tube pulling from plasma membrane spheres, which concur to support the key role of caveolin-1 in building membrane bending rigidity.

To fulfill the recommendations of the reviewers we have modified the manuscript, as discussed below.

**Public Reviews:**

**Reviewer #1 (Public Review):**
Summary:Because of the role of membrane tension in the process, and that caveloae regulate membrane tension, the authors looked at the formation of TEMs in cells depleted of Caveolin1 and Cavin1 (PTRF): They found a higher propensity to form TEMs, spontaneously (a rare event) and after toxin treatment, in both Caveolin 1 and Cavin 1. They show that in both siRNA-Caveolin1 and siRNA-Cavin1 cells, the cytoplasm is thinner. They show that in siCaveolin1 only, the dynamics of opening are different, with notably much larger TEMs. From the dynamic model of opening, they predict that this should be due to a lower bending rigidity of the membrane. They measure the bending rigidity from Cell-generated Giant liposomes and find that the bending rigidity is reduced by approx. 50%.Strengths:They also nicely show that caveolin1 KO mice are more susceptible to death from infections with pathogens that create TEMs.Overall, the paper is well-conducted and nicely written. There are however a few details that should be addressed.

See below modifications brought to the manuscript in response to the Reviewer’s remarks.

**Reviewer #2 (Public Review):**
Summary:The manuscript by Morel et al. aims to identify some potential mechano-regulators of transendothelial cell macro-aperture (TEM). Guided by the recognized role of caveolar invaginations in buffering the membrane tension of cells, the authors focused on caveolin-1 and associated regulator PTRF. They report a comprehensive *in vitro* work based on siRNA knockdown and optical imaging approach complemented with an *in vivo* work on mice, a biophysical assay allowing measurement of the mechanical properties of membranes, and a theoretical analysis inspired by soft matter physics.Strengths:The authors should be complimented for this multi-faceted and rigorous work. The accumulation of pieces of evidence collected from each type of approach makes the conclusion drawn by the authors very convincing, regarding the new role of cavolin-1 as an individual protein instead of the main molecular component of caveolae. On a personal note, I was very impressed by the quality of STORM images (Fig. 2) which are very illuminating and useful, in particular for validating some hypotheses of the theoretical analysis.Weaknesses:While this work pins down the key role of caveolin-1, its mechanism remains to be further investigated. The hypotheses proposed by the authors in the discussions about the link between caveolin and lipids/cholesterol are very plausible though challenging. Even though we may feel slightly frustrated by the absence of data in this direction, the quality and merit of this paper remain.

We thank the reviewer for mentioning the merit of our work which lays the foundations for more molecular mechanistic work on a possible role of lipids/cholesterol in the building of membrane bending rigidity by caveolin-1 and which is currently carried out by some of the authors, and which shows that the question is indeed challenging as indicated by the reviewer. This is now stated in the results section, as suggested (Page 12) :

"To test these predictions, we have treated cells with methyl-beta-cyclodextrin to deplete cholesterol from the plasma membrane and reduce its bending rigidity (47); unfortunately, this treatment affected the cell morphology, which precluded further analysis."

The analogy with dewetting processes drawn to derive the theoretical model is very attractive. However, although part of the model has already been published several times by the same group of authors, the definition of the effective membrane rigidity of a plasma membrane including the underlying actin cortex, was very vague and confusing.

We thank the reviewer for mentioning the importance of defining the terms “membrane bending rigidity” as well as “effective membrane bending rigidity” that is now used and defined in the material and method section in the Physical modelling description (see considerations below), while for the sake of simplicity we use the term “membrane bending rigidity” in the main text, which is now defined in the introduction section : “membrane bending rigidity, i.e. the energy required to locally bend the membrane surface”.

Indeed, in a liposome, a rigorous derivation leads to a relationship between the membrane tension and the variation of the projected area, which are related by the bending rigidity: this relationship is known as the Helfrich’s law. This statistical physics approach is only rigorously valid for a liposome, whereas its application to a cell is questionable due to the presence of cytoskeletal forces acting on the membrane. Nevertheless, application of the Helfrich’s law to cell membranes may be granted on short time scales, before active cell tension regulation takes place (Sens P and Plastino J, 2015 J Phys Condens Matter), especially in cases where cytoskeletal forces play a modest role, such as red blood cells (Helfrich W 1973 Z Naturforsch C). The fact that the cytoskeletal structure and actomyosin contraction are significantly disrupted upon cell intoxication-driven inhibition of the small GTPase RhoA, as shown here for the first time by STORM analysis, supports the applicability of Helfrich’s law to describe TEM opening. Because of the presence of proteins, carbohydrates, and the adhesion of the remaining actin meshwork after toxin treatment, we expect the Helfrich relationship to somewhat differ from the case of a pure lipidic membrane. We account for these effects via an “effective bending rigidity”, a term used in the detailed discussion of the model hypotheses, which corresponds to an effective value describing the relationship between membrane tension and projected area variation in our cells.

The following discussion has been extended and improved in the Physical modeling part of the materials & methods section (Pages 23-24): “κ is the effective bending rigidity of the cell membrane, which quantifies the energy required to bend the membrane. (…). While rigorously derived for a pure lipid membrane, we assumed that Helfrich’s law is applicable to describe the relationship between the effective membrane tension acting on TEMs and the observed projected surface in our cells. We expect Helfrich’s law to be applicable on short time scales, before active cell tension regulation takes place (73), especially in cases where cytoskeletal forces play a modest role, such as for red blood cells (74) or for the highly disrupted cytoskeletal structure of our intoxicated cells. Thus, the parameter κ in Eq. 2 is an effective bending rigidity, whose value may somewhat differ from that of a pure lipid membrane to account for the role played by protein inclusions and the mechanical contribution of the remaining cytoskeletal elements after cell treatment with the toxin”

Here, for the first time, thanks to the STORM analysis, the authors show that HUVECs intoxicated by ExoC3 exhibit a loose and defective cortex with a significantly increased mesh size. This argues in favor of the validity of Helfrich formalism in this context. Nonetheless, there remains a puzzle. Experimentally, several TEMs are visible within one cell. Theoretically, the authors consider a simultaneous opening of several pores and treat them in an additive manner. However, when one pore opens, the tension relaxes and should prevent the opening of subsequent pores. Yet, experimentally, as seen from the beautiful supplementary videos, several pores open one after the other. This would suggest that the tension is not homogeneous within an intoxicated cell or that equilibration times are long. One possibility is that some undegraded actin pieces of the actin cortex may form a barrier that somehow isolates one TEM from a neighboring one.

As pointed by the Reviewer, we expect that membrane tension is neither a purely global nor a purely local parameter. Opening of a TEM will relax membrane tension over a certain distance, not over the whole cell. Moreover, once the TEM closes back, membrane tension will increase again. This spatial and temporal localization of membrane tension relaxation explains that the opening of a first TEM does not preclude the opening of a second one or enlargement of the TEM when the actin cable is cut by laser ablation (20). On the other hand, membrane tension is not a purely local property. Indeed, we observe that when two TEMs enlarge next to each other, their shape becomes anisotropic, as their enlargement is mutually hampered in the region separating them. We account for this interaction by treating TEM membrane relaxation in an additive fashion. We emphasize that this simplified description is used to predict maximum TEM size, corresponding to the time at which TEM interaction is strongest. As the reviewer points out, it would be more questionable to use this additive treatment to predict the likelihood of nucleation of a new TEM, which is not done here.

Accordingly, the Physical modelling part in the materiel and methods has been modified into: “Eq. 2 treats the effect of several simultaneous TEMs in an additive manner. This approximation is used here to predict TEM size, because at maximum opening of simultaneous TEMs their respective membrane relaxation is felt by each other, as it can be inferred from the shape that neighboring TEMs adopt in experiments. This additive treatment would appear less appropriate to describe the likelihood of nucleating a second TEM in the presence of a first one (a calculation that is not performed here), since membrane relaxation by a TEM may not be felt at membrane regions distant from it.”

Could the authors look back at their STORM data and check whether intoxicated cells do not exhibit a bimodal population of mesh sizes and possibly provide a mapping of mesh size at the scale of a cell?

To address the question raised by the Reviewer we decided to plot the whole distribution of mesh sizes in addition to the average value per cell. We did not observe a bimodal distribution but rather a very heterogeneous distribution of mesh size going up to a few microns square in all conditions of siRNA treatments. Moreover, we did not observe a specific pattern in the distribution of mesh size at the scale of the cell, with very large mesh sizes being surrounded by small ones. We also did not observe any specific pattern for the localization of TEM opening, as described in the paper, making the correlation between mesh size and TEM opening difficult.

This following sentence has been added in the results section (Pages 8-9): “Indeed, we observed in cells treated with ExoC3 no specific cellular pattern or bimodal distribution of mesh size between the different siRNA conditions but a rather very heterogeneous distribution of mesh size values that could reach a few square microns in all conditions. ”

In particular, it is quite striking that while bending rigidity of the lipid membrane is expected to set the maximal size of the aperture, most TEMs are well delimited with actin rings before closing. Is it because the surrounding loose actin is pushed back by the rim of the aperture? Could the authors better explain why they do not consider actin as a player in TEM opening?

Actin ring assembly and stiffening is indeed a player in TEM opening, that was investigated in the work by Stefani et al., 2017 Nat comm. Interference of actin ring assembly and stiffening is included in our differential equation describing TEM opening dynamics (second term on the left-hand side of Eq. 3). In some cases, actin ring assembly is the dominant player, such as in TEM opening after laser ablation (ex novo TEM opening/widening). In contrast, here we investigate de novo TEM opening, for which we expect that bending rigidity can be estimated without accounting for actin assembly, as we previously reported (19). Such a bending rigidity estimate (Eq. 5) is obtained by considering two different time scales: the time scale of membrane tension relaxation, governed by bending rigidity, and the time scale of cable assembly, governed by actin dynamics. We expect the first time scale to be shorter, and thus the maximum size of de novo TEMs to be mainly constrained by membrane tension relaxation. Two paragraphs related to the discussion of the different time scales have been added to (1) the discussion section, and (2) to the physical modelling part discussed in the materiel and methods section of the revised manuscript (see below).

The following paragraph has been added in the discussion (Pages 14-15): “Our study shows that membrane rigidity sets the maximal size of TEM aperture, although an actin ring appears before TEM closure (20). Actin ring assembly and stiffening is indeed a player in TEM opening, and it is included in our differential equation describing TEM opening dynamics (Eq. 3). In some configurations, actin ring assembly is the dominant player, such as in TEM opening after laser ablation (ex novo TEM opening), as we previously reported (20). In contrast, here we investigate de novo TEM opening, for which we expect that bending rigidity can be estimated without accounting for actin assembly (19). Such a bending rigidity estimate (Eq. 5) is obtained by considering two different time scales: the time scale of membrane tension relaxation, governed by bending rigidity, and the time scale of cable assembly, governed by actin dynamics. We expect the first-time scale to be shorter, and thus the maximum size of de novo TEMs to be mainly constrained by membrane tension relaxation. However, we cannot rule out that the formation of an actin cable around the TEM before it reaches its maximum size may limit the correct estimation of the bending rigidity.”

The following paragraph has been added in the physical modelling part of the materiel and methods section (Pages 24-25) “A limitation of our theoretical description arises from the use of spatially uniform changes in parameter values to describe differences between experimental conditions, thus assuming spatially uniform effects. However, we cannot exclude the existence of non-uniform effects, such as changes in the size and organization of the remaining actin mesh, which could set local, non-uniform barriers to TEM enlargement in a manner not accounted for by our model.” And “We note that the estimate of κ provided by Eq. 5 is independent of α and thus of actin cable assembly. This simplification arises from membrane tension relaxing over a shorter time scale than actin assembly. Thus, we expect the maximum size of de novo TEMs to be mainly constrained by membrane tension relaxation (19), unlike ex novo TEM enlargement upon laser ablation, for which the dynamics of actin cable assembly control TEM opening (20)”

Instead of delegating to the discussion the possible link between caveolin and lipids as a mechanism for the enhanced bending rigidity provided by caveolin-1, it could be of interest for the readership to insert the attempted (and failed) experiments in the result section. For instance, did the authors try treatment with methyl-beta-cyclodextrin that extracts cholesterol (and disrupts caveolar and clathrin pits) but supposedly keeps the majority of the pool of individual caveolins at the membrane?

As recommended by the reviewer we have added the following sentence (Page 12): “We have treated cells with methyl-beta-cyclodextrin to deplete cholesterol from the plasma membrane and reduce its bending rigidity (47); unfortunately, this treatment affected the cell morphology, which precluded further analysis”

Tether pulling experiments on Plasma membrane spheres (PMS) are real tours de force and the results are quite convincing: a clear difference in bending rigidity is observed in controlled and caveolin knock-out PMS. However, one recurrent concern in these tether-pulling experiments is to be sure that the membrane pulled in the tether has the same composition as the one in the PMS body. The presence of the highly curved neck may impede or slow down membrane proteins from reaching the tether by convective or diffusive motion.

We thank the Reviewer for mentioning the dedicated work accomplished with tether pulling experiments on PMS and for pointing the obtention of convincing results that align well with the hypotheses drawn from the theoretical model thereby allowing us to propose a direct or indirect role of caveolin-1 in the building of membrane rigidity. As pointed out by the reviewer, a concern with tube pulling experiments is related to the dynamics of equilibration of membrane composition between the nanotube and the rest of the membrane. In our experiments, we have waited about 30 seconds after tube pulling and after changing membrane tension. We have checked that after this time, the force remained constant, implying that we have performed experiments of tube pulling from PMS in technical conditions of equilibrium that ensure that lipids and membrane proteins had enough time to reach the tether by convective or diffusive motion.

The revised version of the manuscript now includes the following sentence and a representative example of force vs time plot (Page 12): “We waited about 30 seconds after tube pulling and changing membrane tension and checked that we reached a steady state (Fig. S5), where lipids and membrane proteins had enough time to equilibrate.”

Could the authors propose an experiment to demonstrate that caveolin-1 proteins are not restricted to the body of the PMS and can access to the nanometric tether?

In principle, this could be further checked using cells expressing GFP-caveolin-1 to generate PMS as done in Sinha et al., 2011 and by analyzing a steady protein signal in the tube. This would confirm the equilibration, provided that caveolin-1 is recruited in the nanotube due to mechanical reasons that are now discussed in the discussion section (Pages 13-14) : “Our tube pulling experiments can be discussed along 2 lines. Indeed, since caveolin-1 is inserted in the cytosolic leaflet of the plasma membrane, when a nanotube is pulled towards the exterior of the PMS, we can expect 2 situations depending on the ability of caveolin-1 to deform membranes, which remains to be addressed (24). (i) If Cav1 does not bend membranes, it could be recruited in the nanotube at a density similar to the PMS and our force measurement would reflect the bending rigidity of the PMS membrane. Cav1 could then stiffen membrane either as a stiff inclusion at high density or/and by affecting lipid composition. (ii) If Cav1 bends the membrane, it is expected from caveolae geometry that the curvature in the tube would favor Cav1 exclusion. The force would then reflect the bending rigidity of the membrane depleted of Cav1, which should be the same in both types of experiments (WT and Cav1-depleted conditions) if the lipid composition remains unchanged upon Cav1 depletion. Note that the presence of a very reduced concentration of Cav1 as compared to the plasma membrane has been reported in tunneling nanotubes (TNT) connecting two neighboring cells (51). These TNTs have typical diameters of similar scale than diameters of tubes pulled from PMS. At this stage, we cannot decipher between both properties for Cav1. Considering a direct mechanical role of Cav1, previous studies showed that inclusion of integral proteins in membranes had no impact on bending rigidity, as shown in the bacteriorhodopsin experiment (52), or even decreased membrane rigidity as reported for the Ca2+-ATPase SERCA (53). Previous simulations have also confirmed the softening effect of protein inclusions (54). Nevertheless, our observations could be explained by a high density of stiff inclusions in the plasma membrane (>>10%), which is generally not achievable with the reconstituted membranes. Considering an impact on lipid composition, it is well established that caveolae are enriched with cholesterol, sphingomyelin, and glycosphingolipids, including gangliosides (55,56), which are known to rigidify membranes (57,47). Thus, caveolin-1 might contribute to the enrichment of the plasma membrane with these lipid species. We did not establish experimental conditions allowing us to deplete cholesterol without compromising the shape of HUVECs, which prevented a proper analysis of TEM dynamics. Moreover, a previous attempt to increase TEMs width by softening the membrane through the incorporation of poly-unsaturated acyl chains into phospholipids failed, likely due to homeostatic adaptation of the membrane’s mechanical properties (18). Further studies are now required to establish whether and how caveolin-1 oligomers control membrane mechanical parameters through modulation of lipids organization or content. Caveolin-1 expression may also contribute to plasma membrane stiffening by interacting with membrane-associated components of the cortical cytoskeletal or by structuring ordered lipid domains. Nevertheless, it has been reported that the Young’s modulus of the cell cortex dramatically decreases in ExoC3-treated cells (17) suggesting a small additional contribution of caveolin-1 depletion to membrane softening. This is supported by 2D STORM data showing a dramatic reorganization of actin cytoskeleton in ExoC3-treated cells into a loose F-actin meshwork that is not significantly exacerbated by caveolin-1 depletion. Altogether, our results suggest that the presence of Cav1 stiffens plasma membranes, and that the exact origin of this effect must be further investigated.”

**Author recommendations**

**Reviewer #1 (Recommendations For The Authors):**
Suggestions for improvements:(1) Depletion of both Cavin1 and Caveolin1 increases the density of TEMs. Membrane tension is a critical parameter of the initiation phase of TEMs, its nucleation, and initial enlargement. From the TEM dynamics, the authors should be able to measure membrane tension. The expectation is that in both Caveolin1 and Cavin1 depleted cells, tension is higher (because there is no caveolae), explaining why there are more TEMs.

While we cannot directly measure membrane tension, we can estimate membrane tension variations using our theoretical modeling. As reported in the article, we predict that depleting Caveolin-1 leads to a significant 2-fold increase of membrane tension, which can explain the concomitant increase in the nucleation of TEMs, as the reviewer points out. In contrast, the model predicts no significant increase of membrane tension upon Cavin-1/PTRF depletion, whereas TEM nucleation also increases significantly (but less than upon Caveolin-1 depletion). Altogether, we can explain these results by considering that membrane tension is an important player in TEM nucleation, but not the only one. Notably, we expect cell height to be another important player, as it sets an energy barrier for the basal and apical membranes to meet each other and fuse. Indeed, we report that membrane height is reduced upon depletion Cavin-1, thus explaining the observed increase in TEM nucleation. The importance of reducing cell thickness to increase the TEM opening likelihood is best supported by previous data showing that pushing forces applied on the apical membrane induced the opening of TEMs (Ng et al., 2017 MBoC).

An improved discussion of the parameters controlling TEM nucleation has been included in the discussion of the revised manuscript, as follow (Page 15): “Our study points to underlying mechanisms by which caveolae regulate the frequency of TEM nucleation. Nucleation of TEMs requires the apposition of the basal and apical cell membranes, which is hindered by the intermembrane distance, set by the cell height. Meeting of the two membranes may create an initial precursor tunnel, which needs to be sufficiently big to enlarge into an observable TEM, instead of simply closing back. The size of the minimal precursor tunnel required to give rise to a TEM increases with membrane bending rigidity and decreases with membrane tension (19). Silencing cavin-1 or caveolin-1 both lead to a decrease in cell height, thus favoring the likelihood of precursor tunnel nucleation. While silencing cavin-1 has no significant impact on either membrane tension or bending rigidity, silencing caveolin results in both an increase of membrane tension and a decrease of bending rigidity, which results in a decrease in the required minimal radius of the precursor tunnel, thus further favoring TEM nucleation. Overall, our results offer a consistent picture of the physical mechanisms by which caveolae modulate TEM nucleation.”

(2) In Figure 2B, the authors state that there is no significant difference in the actin mesh size while I see a clear higher average value and distribution in siCAV1+. This seems to correlate with the differences in TEM maximal sizes. How can the authors completely exclude that the actin organisation is not in part responsible for the larger TEMs observed in siCAV1 cells?

In our theoretical modeling of TEM opening dynamics, all differences between conditions are described by changes in what we consider as “effective” parameter values. Thus, changes in actin organization may induce a change in the "effective bending rigidity" parameter controlling membrane tension relaxation. A limitation of such a description is that all changes are assumed to be spatially uniform. However, it is possible that changes in actin mesh size and organization set local barriers to TEM enlargement in a way that would not be appropriately described by our model. While our current modeling appears to provide a consistent interpretation of our observations, we cannot completely exclude the existence of such local effects.

This limitation of our current interpretation is now mentioned in the following paragraph, which has been added in the physical modelling part of the materiel and methods section (Page 24) : “A limitation of our theoretical description arises from the use of spatially uniform changes in parameter values to describe differences between experimental conditions, thus assuming spatially uniform effects. However, we cannot exclude the existence of non-uniform effects, such as changes in the size and organization of the remaining actin mesh, which could set local, non-uniform barriers to TEM enlargement in a manner not accounted for by our model.”

(3) It would be nice to see the results of Table 1 (in particular the thickness of cells) in a Bar plot.

The experimental values of cell volumes and areas are reported in bar plots of Fig. 3C and 3D. In contrast, we chose not to depict values of cell eight in bar plots considering that these values were calculated from mean values of cell areas and volumes reported in Fig. 3C and 3D, i.e. rough division of volumes over areas, with error propagation. Since the volume and areas are not performed on the same set of cells, it is not possible to divide the repeats one by one and to provide cell numbers, which are key parameters to perform statistical tests.

(4) There are two reasons why Caveolin1 could change the bending rigidity. First, because it makes the membrane stiffer, or because the presence of caveolin1 (that binds to cholesterol) in the plasma membrane changes the lipid composition. It would be nice if the authors could provide some lipidomics analysis to see if there is a lipid change in siCAV1 cells.

We thank the reviewer for pointing the importance of clarifying the hypotheses regarding a direct or indirect role of caveolin-1 in membrane bending rigidity which might be related to changes in membrane lipid composition especially cholesterol and sphingomyelin. We have modified the discussion section to integrate this point. The lipidomic approach is certainly interesting to address the question of the role of caveolin-1 in building membrane bending rigidity. Indeed, some of the authors have addressed the specific questions related to Cav-1 spontaneous curvature and its effect on the lipid composition of the plasma membrane in two separate manuscripts (in preparation). They represent comprehensive studies by themselves that will provide mechanistic insights on how caveolin-1 builds membrane bending rigidity and as follow up of the present manuscript which reports the importance of the regulation of membrane rigidity in cell biology and during infectious processes.

**Reviewer #2 (Recommendations For The Authors):**
The paper is nicely written and the results are convincing. The three main comments and questions from the Public Review do not necessarily call for new experiments. However, clarifications are required. This work can be very useful. Better not to leave any difficulty or weakly justified hypothesis under the carpet.

To fulfill with the reviewer comments, we have improved the discussion regarding the hypothesizes which can be drawn about of a direct versus indirect mechanistic role of caveolin-1 in the regulation of effective membrane bending rigidity and which might be related to changes in membrane lipid composition or via regulation of the cytoskeleton, which we cannot exclude.

Minor correction: in the abstract: replace "the enhanced nucleation" with "the enhanced occurrence of nucleation events".

The abstract has been changed accordingly : “The enhanced occurrence of TEM nucleation events correlates with a reduction of cell height, …”